# Functional connectivity-based attractor dynamics of the human brain in rest, task, and disease

**Robert Englert[1,2], Balint Kincses[1,3], Raviteja Kotikalapudi[1,3], Giuseppe Gallitto[1,3], Jialin Li[1,3,4], Kevin Hoffschlag[1,3], Choong-Wan Woo[5,6], Tor D Wager[7], Dagmar Timmann[1,3], Ulrike Bingel[1,3], Tamas Spisak[1,2]***

[1]Center for Translational Neuro- and Behavioral Sciences (C-TNBS), University Medicine Essen, Essen, Germany; [2]Department of Diagnostic and Interventional Radiology and Neuroradiology, University Medicine Essen, Essen, Germany; [3]Department of Neurology, University Medicine Essen, Essen, Germany; [4]Max Planck School of Cognition, Leipzig, Germany; [5]Center for Neuroscience Imaging Research, Institute for Basic Science, Suwon, Republic of Korea; [6]Department of Biomedical Engineering, Sungkyunkwan University, Suwon, Republic of Korea; [7]Department of Psychological and Brain Sciences, Dartmouth College, Hanover, United States

**\*For correspondence:**
tamas.spisak@uk-essen.de

**Competing interest:** The authors declare that no competing interests exist.

## eLife Assessment

This study presents a **valuable** approach for revealing large-scale brain attractor dynamics during resting states, task processing, and disease conditions using insights from Hopfield neural networks. The evidence supporting the findings is **convincing** across the many datasets analysed. The work will be of broad interest to neuroscientists using neuroimaging data with interest in computational modelling of brain activity.

**Abstract** Functional brain connectivity has been instrumental in uncovering the large-scale organization of the brain and its relation to various behavioral and clinical phenotypes. Understanding how this functional architecture relates to the brain's dynamic activity repertoire is an essential next step towards interpretable generative models of brain function. We propose functional connectivity-based Attractor Neural Networks (fcANNs), a theoretically inspired model of macro-scale brain dynamics, simulating recurrent activity flow among brain regions based on first principles of self-organization. In the fcANN framework, brain dynamics are understood in relation to attractor states; neurobiologically meaningful activity configurations that minimize the free energy of the system. We provide the first evidence that large-scale brain attractors - as reconstructed by fcANNs - exhibit an approximately orthogonal organization, which is a signature of the self-orthogonalization mechanism of the underlying theoretical framework of free-energy-minimizing attractor networks. Analyses of seven distinct human neuroimaging datasets demonstrate that fcANNs can accurately reconstruct and predict brain dynamics under a wide range of conditions, including resting and task states, and brain disorders. By establishing a formal link between connectivity and activity, fcANNs offer a simple and interpretable computational alternative to conventional descriptive analyses.

## Introduction

Brain function is characterized by the continuous activation and deactivation of anatomically distributed neuronal populations (*Buzsaki, 2006*). Irrespective of the presence or absence of explicit stimuli, brain regions appear to work in concert, giving rise to rich and spatiotemporally complex fluctuations (*Bassett and Sporns, 2017*). These fluctuations are not random (*Liu and Duyn, 2013*; *Zalesky et al., 2014*); they organize around large-scale gradients (*Margulies et al., 2016*; *Huntenburg et al., 2018*) and exhibit quasi-periodic properties, with a limited number of recurring patterns often termed as 'brain substates' (*Greene et al., 2023*; *Kringelbach and Deco, 2020*; *Vidaurre et al., 2017*; *Liu and Duyn, 2013*). A wide variety of descriptive techniques have been previously employed to characterize whole-brain dynamics (*Smith et al., 2012*; *Vidaurre et al., 2017*; *Liu and Duyn, 2013*; *Chen et al., 2018*). These efforts have provided accumulating evidence not only for the existence of dynamic brain substates but also for their clinical significance (*Hutchison et al., 2013*; *Barttfeld et al., 2015*; *van der Meer et al., 2020*). However, the underlying driving forces remain elusive due to the descriptive nature of such studies.

Conventional computational approaches attempt to solve this puzzle by going all the way down to the biophysical properties of single neurons and aim to construct a model of larger neural populations, or even the entire brain (*Breakspear, 2017*). These approaches have shown numerous successful applications (*Murray et al., 2018*; *Kriegeskorte and Douglas, 2018*; *Heinz et al., 2019*). However, such models need to estimate a vast number of neurobiologically motivated free parameters to fit the data. This hampers their ability to effectively bridge the gap between explanations at the level of single neurons and the complexity of behavior (*Breakspear, 2017*). Recent efforts using coarse-grained brain network models (*Schirner et al., 2022*; *Schiff et al., 1994*; *Papadopoulos et al., 2017*; *Seguin et al., 2023*) and linear network control theory (*Chiêm et al., 2021*; *Scheid et al., 2021*; *Gu et al., 2015*) opted to trade biophysical fidelity to phenomenological validity. Such models have provided insights into some of the inherent key characteristics of the brain as a dynamic system; for instance, the importance of stable patterns, the attractor states, in governing brain dynamics (*Deco et al., 2012*; *Golos et al., 2015*; *Hansen et al., 2015*). While attractor networks have become established models of micro-scale canonical brain circuits in the last four decades (*Khona and Fiete, 2022*), these studies suggest that attractor dynamics are essential characteristics of macro-scale brain dynamics as well (*Poerio and Karapanagiotidis, 2025*). Attractor networks, however, come in many flavors, and the specific forms and behaviors of these networks are heavily influenced by the chosen inference and learning rules, making it unclear which variety should be in focus when modeling brain dynamics. Given that the brain showcases not only multiple signatures of attractor dynamics but also the ability to evolve and adapt through self-organization (i.e. in the absence of any centralized control), investigating attractor models from the point of view of self-organization may be key to narrow down the set of viable models.

In our recent theoretical work (*Spisak and Friston, 2025*), we identified the class of attractor networks that emerge from first principles of self-organization, as articulated by the Free Energy Principle (FEP) (*Friston, 2010*; *Friston et al., 2023*), and identified the emergent inference and learning rules guiding the dynamics of such systems. This theoretical framework reveals that the minimization of variational free energy locally, e.g., by individual network nodes, gives rise to a dual dynamic: simultaneous inference (updating activity) and learning (optimizing connectivity). The emergent inference process in these systems is equivalent to local Bayesian update dynamics for the individual network nodes, homologous to the stochastic relaxation observed in conventional Boltzmann neural network architectures (e.g. stochastic Hopfield networks, *Hopfield, 1982*; *Koiran, 1994*), and in line with the empirical observation that activity in the brain 'flows' following similar dynamics (*Cole et al., 2016*; *Sanchez-Romero et al., 2023*; *Cole, 2024*). Importantly, in this framework, attractor states are not simply an epiphenomenon of collective dynamics, but serve as global priors in the Bayesian sense, that get combined with the current activity configuration so that the updated activity samples from the posterior (akin to a Markov-Chain Monte Carlo (MCMC) sampling process).

Learning, on the other hand, emerges in this framework in the form of a distinctive coupling plasticity, a local, incremental learning rule, that continuously adjusts coupling weights to preserve low free energy in anticipation of future sensory encounters following a contrastive predictive coding scheme (*Millidge et al., 2022*), effectively implementing action selection in the active inference sense (*Friston et al., 2016*). Importantly, the learning dynamics emerging in our theoretical framework provide a

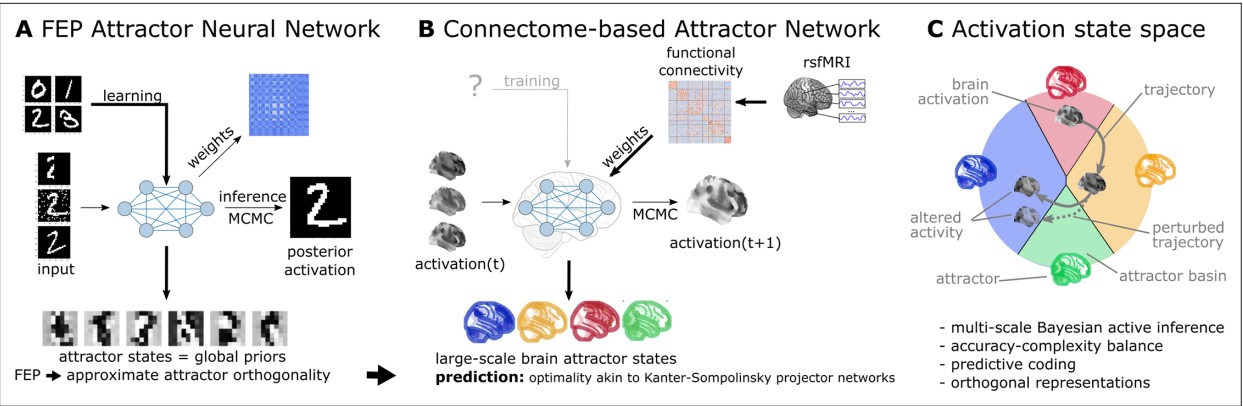

**Figure 1.** Functional connectivity-based attractor neural networks as models of macro-scale brain dynamics. (**A**) Free-energy-minimizing artificial neural networks (***Spisak and Friston, 2025***) are a form of recurrent stochastic artificial neural networks that, similarly to classical Hopfield networks (***Hopfield, 1982***; ***Koiran, 1994***), can serve as content-addressable ('associative') memory systems. More generally, through the learning rule emerging from local free-energy minimization, the weights of these networks will encode a global internal model of the external world. The priors of this internal generative model are represented by the attractor states of the network that, as a special consequence of free-energy minimization, will tend to be orthogonal to each other. During stochastic inference (local free-energy minimization), the network samples from the posterior that combines these priors with the previous brain substates (also encompassing incoming stimuli), akin to Markov chain Monte Carlo (MCMC) sampling. (**B**) In accordance with this theoretical framework, we consider regions of the brain as nodes of a free-energy-minimizing artificial neural network. Instead of initializing the network with the structural wiring of the brain or training it to solve specific tasks, we set its weights empirically, using information about the interregional 'activity flow' across regions, as estimated via functional brain connectivity. Applying the inference rule of our framework - which displays strong analogies with the relaxation rule of Hopfield networks and the activity flow principle that links activity to connectivity in brain networks - results in a generative computational model of macro-scale brain dynamics that we term a functional connectivity-based (stochastic) attractor neural network (fcANN). (**C**) The proposed computational framework assigns a free energy level, a probability density and a 'trajectory of least action' towards an attractor state to any brain activation pattern and predicts changes of the corresponding dynamics in response to alterations in activity and/or connectivity. The theoretical framework underlying the fcANNs - based on the assumption that the brain operates as a free energy minimizing attractor network - draws formal links between attractor dynamics and multi-level Bayesian active inference.

strong, testable hypothesis: if the brain operates as a free energy minimizing attractor network, its large-scale attractors should be approximately orthogonal to each other. This is not a general property of all recurrent (attractor) neural networks, but a direct consequence of free energy minimization, shown both mathematically and with simulations in ***Spisak and Friston, 2025***. As our theoretical framework, by design, embraces multiple valid levels of description (through coarse-graining), it is well-suited to serve as a basis for a computational model of large-scale brain dynamics.

In the present work, we translate the results of this novel theoretical framework into a computational model of macro-scale brain dynamics and deploy a diverse set of experimental, clinical, and meta-analytic studies to perform an initial investigation of several of its predictions. We start by showing that, if large-scale brain dynamics evolve and organize according to the emergent rules of this framework, the corresponding attractor model can be effectively approximated from functional connectivity data, as measured with resting-state fMRI. Based on the network topology spanned by functional connectivity, our model assigns a free energy level for any arbitrary activation pattern and determines a 'trajectory of least action' towards one of a finite number of attractor states that minimize this energy (***Figure 1***). We then perform an initial investigation of the robustness and biological plausibility of the attractor states of the reconstructed network and whether it is able to reproduce various characteristics of resting-state brain dynamics. Importantly, we directly test the framework's prediction on the emergence of (approximately) orthogonal attractor states. Capitalizing on the generative nature of our model, we also demonstrate how it can capture and potentially explain the effects of various perturbations and alterations of these dynamics, from task-induced activity to changes related to brain disorders.

## Theoretical background
### Free energy minimizing artificial neural networks
The computational model at the heart of this work is a direct implementation of the inference dynamics that emerge from a recently proposed theoretical framework of free energy minimizing

(self-orthogonalizing) attractor networks (*Spisak and Friston, 2025*) (FEP-ANNs). FEP-ANNs are a class of attractor neural networks that emerge from the FEP (*Friston, 2010*; *Friston et al., 2023*). The FEP posits that any self-organizing random dynamical system that maintains its integrity over time (i.e. has a steady-state distribution and a statistical separation from its environment) must act in a way that minimizes its variational free energy. FEP-ANNs apply the FEP recursively. We assume a network of $N$ units, where each unit is represented by a single continuous-valued state variable $\sigma_i \in [-1, 1]$, so that the activity of the network is described by a vector of states $\sigma = (\sigma_1, \sigma_2, \ldots, \sigma_N)$ and these states are conditionally independent of each other, given boundary states that realize the necessary statistical separation between them (corresponding to a complex Markov blanket structure in the FEP terminology). When assuming $\sigma_i$ states that follow a continuous Bernoulli (a.k.a. truncated exponential: $p(\sigma_i) \propto e^{\kappa_i \sigma_i}$) distribution (parameterized by the single parameter $\kappa_i$) and deterministic couplings $J$, the steady-state distribution can be expressed as:

$$p^*(\sigma) \propto \exp\left(\sum_i b_i \sigma_i + \frac{1}{2} \sum_{i,j} J_{ij}^S \sigma_i \sigma_j\right) \tag{1}$$

where $b_i$ represents the local evidence or bias for each unit $i$ (e.g. external input or intrinsic excitability of a brain region), $J^S = \frac{1}{2}(J + J^\top)$ is the symmetric component of the coupling weights between units $i$ and $j$, and $\beta$ is an inverse temperature or precision parameter. Note that while this steady-state distribution has the same functional form as continuous-state Boltzmann machines or stochastic Hopfield networks, the true coupling weights $J$ do not have to be symmetric as usually assumed in those architectures. Asymmetric couplings break detailed balance, meaning that $p^*$ is no longer an equilibrium distribution. However, the antisymmetric component $J^A = \frac{1}{2}(J - J^\top)$ does not contribute to the steady-state distribution $p^*$ as it only induces circulating (solenoidal) flow in the state space which is tangential to the level sets of $p^*$. Thus, while the overall framework can describe general attractor networks with asymmetric couplings and non-equilibrium steady states (NESS), it also implies that knowing only the symmetric component of the coupling weights is sufficient to reconstruct the steady-state distribution $p^*$ of the underlying system. This is a highly useful property for the purposes of the present study, where the couplings are reconstructed from resting state fMRI data, without any explicit information about the directionality of functional connections. For a detailed derivation of the steady-state distribution, see *Spisak and Friston, 2025* and Supplementary Information 1.

Knowing the steady-state distribution of a free-energy-minimizing attractor network, we can derive two types of emergent dynamics from the single imperative of free energy minimization: inference and learning.

## Inference: stochastic relaxation dynamics

Inference arises from minimizing free energy with respect to the states $\sigma$. For a single unit, this yields a local update rule homologous to the relaxation dynamics in Hopfield networks:

$$E_q[\sigma_i] = L(b_q) = L\left(\underbrace{\underbrace{b_i}_{\text{bias}} + \underbrace{\sum_{j \neq i} J_{ij} \sigma_j}_{\text{local potential}}}_{\text{sigmoid (Langevin)}}\right) \tag{2}$$

where $L$ is a sigmoidal activation function (a Langevin function in our case). This rule dictates that each unit updates its activity stochastically, based on a weighted sum of the activity of other units, plus its own intrinsic bias. See (*Spisak and Friston, 2025*) and Appendix 3 for a detailed derivation of the inference dynamics.

Note that the rule is expressed in terms of the expected value of the state $\sigma_i$, which is a stochastic quantity. However, in the limiting case of symmetric couplings (which is the case throughout the present study) and least-action dynamics (i.e. no noise), this update rule reduces to the classical relaxation dynamics of (continuous-state) Hopfield networks. In the present study, we use both the deterministic

('least action') and stochastic variants of the inference rule. The former identifies attractor states; the latter serves as a generative model for large-scale, multistable brain dynamics.

In the present study, we make the simplifying assumption that all nodes have zero bias (b = 0). Furthermore, we allow investigating different scaling factors for the $J$ couplings matrix (given the uncertainties around the magnitude of association in the functional connectome) by introducing a 'scaling factor' $\beta$. This leads to the following update rule:

$$\sigma_i^{(t+1)} = L\left(\beta \sum_{j\neq i} J_{ij}\sigma_j^{(t)}\right) + \text{noise} \tag{3}$$

The scaling factor $\beta$ is analogous to the inverse temperature parameter known in Hopfield networks and Boltzmann machines.

In the basis framework (*Spisak and Friston, 2025*), inference is a gradient descent on the variational free energy landscape with respect to the states $\sigma$ and can be interpreted as a form of approximate Bayesian inference, where the expected value of the state $\sigma_i$ is interpreted as the posterior mean given the attractor states currently encoded in the network (serving as a macro-scale prior) and the previous state, including external inputs (serving as likelihood in the Bayesian sense). The stochastic update, therefore, is equivalent to a Markov chain Monte Carlo (MCMC) sampling from this posterior distribution. The inverse temperature parameter β, in this regard, can be interpreted as the precision of the prior encoded in $J$. This is easy to conceptualize by considering the limiting case of infinite precision, where the system simplifies to a binary-state Hopfield network ($\beta \rightarrow \infty$, $L(\beta u_i) \rightarrow \text{sign}(u_i)$ on $[-1, 1]$) that directly and deterministically converges to the (infinite-precision) prior, completely overriding the Bayesian likelihood (i.e. network input).

## Free energy-minimizing attractor networks as a model of large-scale brain dynamics

Taken together, the novel framework of free energy-minimizing attractor networks not only motivates the use of a specific, emergent class of attractor networks as models for large-scale brain dynamics, but also provides a formal connection between these dynamics and Bayesian inference. The present study leverages this theoretical foundation. We aim to model large-scale brain dynamics as a free energy-minimizing attractor network. According to our framework, such networks can be reconstructed from the activation time-series data measured in their nodes. Specifically, the weight matrix of the attractor network can be reconstructed as the negative inverse covariance matrix of the regional activation time series: $J = -\Lambda = -\Sigma^{-1}$, where $\Sigma$ is the covariance matrix of the activation time series in all regions, and $\Lambda$ is the precision matrix. For a detailed derivation, see Appendix 5. Note that this approach can naturally be reduced to different 'coarse-grainings' of the system, by pooling network nodes with similar functional properties. In the case of resting-state fMRI data, this corresponds to pooling network nodes into functional parcels. Drawing upon concepts, such as the center manifold theorem (*Wagner, 1989*), it is posited that rapid, fine-grained dynamics at lower descriptive levels converge to lower-dimensional manifolds, upon which the system evolves via slower processes at coarser scales. It has been previously argued (*Medrano et al., 2024*) that the temporal and spatial scales of fMRI data happen to align relatively well with the characteristic scales corresponding to meaningful large-scale 'coarse-grainings' of brain dynamics.

Thus, we can reconstruct FEP-ANNs from functional connectivity data simply by considering the functional connectome (inverse covariance or partial correlation) as the coupling weights between the nodes of the network, which themselves correspond to brain regions (as defined by the chosen functional brain parcellation). We refer to such network models as functional connectivity-based attractor neural networks - **fcANN**s for short.

Having estimates of the weight matrix $J$ of the attractor network, we can now rely on the deterministic and stochastic versions of the inference procedure (*Equation 3*) in order to investigate this system. Running the deterministic update to a uniformly drawn sample of initial states, we can identify all attractor states of the network. The stochastic update, on the other hand, can be used to sample from the posterior distribution of the activity states, and thus serves as a generative computational model of the brain dynamics.

## Testable predictions of the theoretical framework

### Self-orthogonalization as a signature of free energy attractor networks

So far, we have only discussed free energy minimization in terms of the activity of the nodes of the network. However, free energy minimization also gives rise to a specific learning rule for the couplings **J** of the network. This learning rule is a specific local, incremental, contrastive (predictive coding-based) plasticity rule to adjust connection strengths:

$$\Delta \mathbf{J}_{ij} \propto \underbrace{\sigma_i \sigma_j}_{\text{observed correlation (Hebbian)}} - \underbrace{\mathrm{L}\left(b_i + \sum_{k \neq i} J_{ik}\, \sigma_k\right)\sigma_j}_{\text{predicted correlation (anti-Hebbian)}} \tag{4}$$

A detailed derivation of the learning dynamics can be found in *Spisak and Friston, 2025* and Appendix 3. In the present work, we do not implement this learning rule in our computational model, as the coupling weights **J** are reconstructed directly from the empirical fMRI activation time series data.

However, this specific learning rule has an important implication for the attractor states of the FEP-ANN: it will naturally drive them towards (approximate) orthogonality during learning. For a mathematical motivation of the mechanisms underlying this important property, termed self-orthogonalization, see *Spisak and Friston, 2025* and Appendix 4. Self-orthogonalization is far from being a generic property of all attractor networks (and it is also not a consequence of the above-formulated inference dynamics). It has, however, remarkable implications for the computational efficiency of the network and the robustness of its representations. Attractor networks with orthogonal attractor states, often termed the Kanter-Sompolinsky projector neural network (*Kanter and Sompolinsky, 1987*), are the computationally most efficient varieties of general attractor networks, with maximal memory capacity and perfect memory recall (without error). Importantly, in Kanter-Sompolinsky projector networks, **the eigenvectors of the coupling matrix and the attractors become equivalent**, providing an important signature for detecting such networks in empirical data.

Importantly, in the present study, we reconstruct attractor networks from functional connectivity data (fcANNs) without relying on the learning rule of the FEP-ANN framework (*Equation 4*), which imposes orthogonality on the attractors. Thus, if, in these empirically reconstructed fcANNs, an alignment between the eigenvectors of the coupling matrix and the attractors is observed, it can be considered strong evidence that the system approximates a Kanter–Sompolinsky projector network. As FEP-ANNs - together with some other, related models (e.g. 'dreaming neural networks,' *Hopfield et al., 1983*; *Plakhov Yu, 1994*; *Dotsenko and Tirozzi, 1991*; *Fachechi et al., 2019*) - provide a plausible and mathematically rigorous mechanistic model for the emergence of architectures approximating Kanter–Sompolinsky projector networks through biologically plausible local learning rules, this alignment between the eigenvectors of the coupling matrix and the attractors can be considered a signature of an underlying FEP-ANN. We will directly test this prediction in the present study, by investigating the orthogonality of the attractor states of the fcANN model reconstructed from empirical fMRI data.

### Convergence, multistability, biological plausibility and prediction capacity

Beyond (approximate) attractor orthogonality, our framework provides additional testable predictions. If the functional connectome can indeed be considered a proxy for the coupling weights **J** of an underlying attractor network, we can expect that (i) the reconstructed fcANN model will exhibit multiple stable attractor states, with large basins and biologically plausible spatial patterns, (ii) the relaxation dynamics of the reconstructed model will display fast convergence to attractor states, and (iii) the stochastic relaxation dynamics yield an efficient generative model of the empirical resting-state brain dynamics as well as perturbations thereof caused either by external inputs (stimulations and tasks) or pathologies.

### Research questions

We have structured the present work around 7 research questions we address in the present study:

## Q1 - Is the brain an approximate K-S projector ANN (FEP-ANN prediction)?

We test whether fcANN-derived brain attractor states closely resemble the eigenvectors of the functional connectome matrix, in contrast to null models based on temporally phase-randomized time series data (preserving the frequency spectrum and the temporal autocorrelation of data, but destroying conditional dependencies across regions), denoted as **NM1**. Furthermore, in a supplementary analysis, we quantify the similarity of the functional connectome to the weights of an optimal Kanter–Sompolinsky (K-S) network with the same eigenvectors. The similarity (cosine similarity) is contrasted against repeating the same approach on permuted coupling matrices (but retaining symmetry, **NM2**).

## Q2 - Is the functional connectome well-suited to function as an attractor network?

We contrast the convergence properties of fcANN deterministic relaxation dynamics with null models with permuted coupling weights (preserving symmetry, sparsity, and weight distributions, destroying topological structure) NM2.

## Q3 - What are the optimal parameters for the fcANN model?

The number of attractor states is a function of the inverse temperature parameter $\beta$. For simplicity, we fix $\beta = 0.04$ (four attractor states) in the current analysis. We perform a rough optimization of the noise parameter $\epsilon$ by benchmarking the fcANN's ability to capture non-Gaussian conditional distributions in the data. This is benchmarked by computing a Wasserstein distance between the distributions of empirical and simulated data and contrasting it to the null model of a multivariate normal distribution with covariance matched to that of the empirical data (**NM3**, representing the case of Gaussian-only conditionals).

## Q4 - Do fcANNs display biologically plausible attractor states?

We qualitatively demonstrate that attractor states obtained with different inverse temperature parameters $\beta$ and different noise levels ($\epsilon$) exhibit large basins and that these attractor states exhibit spatial patterns consistent with known large-scale brain systems.

## Q5 - Can fcANNs reproduce the characteristics of resting-state brain activity?

We compare how well fcANN attractor states explain variance in unseen (in- and out-of-sample) empirical time series data, relative to the principal components of the empirical data itself. Statistical significance is evaluated via bootstrapping. Furthermore, we compare various characteristics (state occupancy, distribution, temporal trajectory) of the data generated by fcANNs via stochastic updates to empirical resting-state data. As null models, we use covariance-matched multivariate normal distributions (NM3).

**Table 1.** Null models applied in the present study.

| Short name | Brief description | Invariant to | Destroys |
|---|---|---|---|
| NM1 Temporal phase randomization | Phase-randomize time series data independently for each region; recalculate connectivity. | Time-series power spectrum and autocorrelation | Conditional dependencies across regions |
| NM2 Symmetry-preserving matrix permutation | Shuffle off-diagonal entries of $J$ while keeping symmetry | Weight distribution and symmetry | Topological structure, clusteredness |
| NM3 Covariance-matched Gaussian | Draw time frames from a multivariate normal with covariance equal to the functional connectome's covariance | Gaussian conditionals | Nonlinear and non-Gaussian conditionals, temporal autocorrelation |
| NM4 Temporal order permutation | Randomly permute time-frame order within runs; used for flow analyses | Spatial autocorrelation | Temporal autocorrelation |
| NM5 Condition shuffling | Permute condition labels, either within participant (e.g. pain vs. rest; up- vs. down-regulation) or between participant (shuffle patient vs. control labels) | Marginal distributions and overall data structure | Condition-specific associations and effects |

**Table 2.** Research questions, methodological approaches, and the corresponding null models.

| Research question | Methodological approach | Null model |
|---|---|---|
| **Q1**. Is the brain an approximate K-S projector ANN (FEP-ANN prediction)? | Compare eigenvectors of the coupling matrix with attractor states | NM1-2 |
| **Q2**. Is the functional connectome well-suited to function as an attractor network? | Measure iterations to convergence in deterministic relaxation | NM2 |
| **Q3**. What are the optimal parameters for the fcANN model? | We fix $\beta$ = 0.04 (four attractor states) for simplicity. We perform a rough optimization of the noise parameter $\epsilon$ in stochastic relaxation to match empirical data distribution. | NM3 |
| **Q4**. Do fcANNs display biological plausible attractor states? | Identify attractor states, report basin sizes, and assess spatial patterns with different inverse temperature parameters and noise levels | Qualitative |
| **Q5**. Can fcANNs reproduce the characteristics of resting-state brain activity? | Compare stochastic dynamics (state occupancy, distribution, temporal trajectory) with empirical resting state data | NM3-4 |
| **Q6**. Can resting-state fMRI-based fcANNs predict large-scale brain dynamics elicited by tasks or stimuli? | Contrast pain vs. rest dynamics with data generated by fcANNs and pain-associated control signal | NM5 |
| **Q7**. Can resting-state fMRI-based fcANNs predict altered brain dynamics in clinical populations? | Contrast autism spectrum disorder patients vs. typically developing control participants' observed brain dynamics with data generated by fcANNs initialized with the respective functional connectomes | NM5 |

## Q6 - Can resting-state fMRI-based fcANNs predict large-scale brain dynamics elicited by tasks or stimuli?

We test whether fcANNs initialized from resting-state functional connectomes and perturbed with weak, condition-specific control signals predict task-evoked large-scale dynamics (pain vs. rest; up- vs. down-regulation). We compare simulated and empirical differences on the fcANN projection and flow fields during stochastic updates. As a null model, we use condition-label shuffling (NM5).

## Q7 - Can resting-state fMRI-based fcANNs predict altered brain dynamics in clinical populations?

We test whether fcANNs initialized with group-level resting-state connectomes from autism spectrum disorder (ASD) patients and typically developing controls (TDC) predict observed group differences in dynamics (state occupancy, attractor-basin activations, flow fields). We compare fcANN-generated dynamics between ASD- and TDC-initialized models and evaluate similarity to empirical contrasts. As a null model, we use group-label shuffling (NM5).

For a summary of null modeling approaches and research questions, see *Tables 1 and 2*.

## Results

### Functional connectivity-based attractor networks (fcANNs) as a model of brain dynamics

First, we constructed a functional connectivity-based attractor network (fcANN) based on resting-state fMRI data in a sample of n=41 healthy young participants (study 1). Details are described in the Methods. In brief, we estimated interregional activity flow (*Cole et al., 2016*; *Ito et al., 2017*) as the study-level average of regularized partial correlations among the resting-state fMRI time series of m=122 functional parcels of the BASC brain atlas (see Methods for details). We then used the standardized functional connectome as the $J_{ij}$ weights of a fully connected recurrent fcANN model, see Methods.

Next, we applied the deterministic relaxation procedure to a large number of random initializations (n=100000) to obtain all possible attractor states of the fcANN in study 1 (*Figure 2A*). Consistent with theoretical expectations, we observed that increasing the inverse temperature parameter β led to an increasing number of attractor states (*Figure 2*, *Figure 2—figure supplement 3*, left, *Figure 2—figure supplement 1*), appearing in symmetric pairs (i.e. $\sigma_i^{(1)} = -\sigma_i^{(2)}$, see *Figure 2G*).

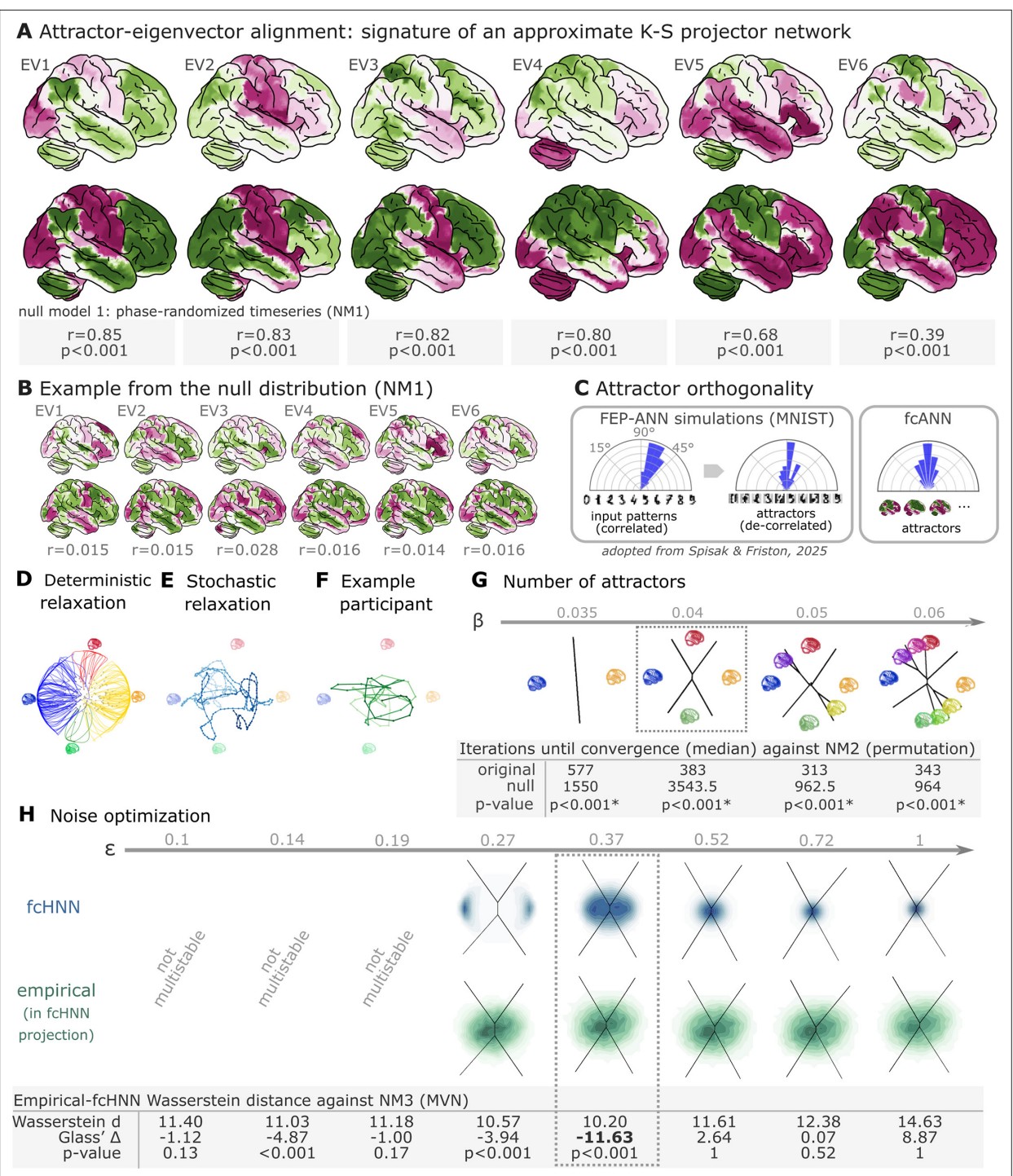

**Figure 2.** Attractor states and state-space dynamics of connectome-based Hopfield networks. (**A**) Leading eigenvectors of the empirical coupling matrix $J$ (upper in each pair) closely match functional connectivity-based attractor network (fcANN) attractor states (lower in each pair). Numbers under each pair report Pearson correlation and two-sided p-values based on 1000 surrogate data realizations, generated by phase-randomizing the true time series and recomputing the connectivity matrix. For the comprehensive results of the eigenvector-attractor alignment analysis (including a supplementary analysis on weight similarity to the analogous Kanter-Sompolinsky projector network) see **Figure 2—figure supplement 2**. (**B**) Example matches from a single permutation of the permutation-based null distribution. For each symmetry-preserving permutation of $J$, we recomputed the corresponding eigenvectors and attractors and re-matched them. The maps are visibly mismatched and correlations are near zero, illustrating the null against which the empirical correlations in panel **A** are evaluated. (**C**) Left panel: Free-energy-minimizing attractor networks have been shown to establish approximately orthogonal attractor states (right), even when presented with correlated patterns (left, adapted from **Spisak and Friston, 2025**). fcANN analysis reveals that the brain also exhibits approximately orthogonal attractors. On all three polar plots, pairwise angles between attractor states are shown. Angles

*Figure 2 continued*

concentrating around 90° in the empirical fcANN are consistent with predictions of free-energy-minimizing (Kanter-Sompolinsky-like) networks. (Note, however, that in high-dimensional spaces, random vectors would also tend to be approximately orthogonal.) (**D**) The fcANN of study 1 seeded with real activation maps (gray dots) of an example participant. All activation maps converge to one of the four attractor states during the deterministic relaxation procedure (without noise), and the system reaches equilibrium. Trajectories are colored by attractor state. (**E**) Illustration of the stochastic relaxation procedure in the same fcANN model, seeded from a single starting point (activation pattern). With stochastic relaxation, the system no longer converges to an attractor state, but instead traverses the state space in a way restricted by the topology of the connectome and the 'gravitational pull' of the attractor states. The shade of the trajectory changes with increasing number of iterations. The trajectory is smoothed with a moving average over 10 iterations for visualization purposes. (**F**) Real resting state fMRI data of an example participant from study 1, plotted on the fcANN projection. The shade of the trajectory changes with an increasing number of iterations. The trajectory is smoothed with a moving average over 10 iterations for visualization purposes. (**G**) Consistent with theoretical expectations, we observed that increasing the inverse temperature parameter $\beta$ led to an increasing number of attractor states, emerging in a nested fashion (i.e. the basin of a new attractor state is fully contained within the basin of a previous one). When contrasting the functional connectome-based attractor neural network (ANN) with a null model based on symmetry-retaining permuted variations of the connectome (NM2), we found that the topology of the original (unpermuted) functional brain connectome makes it significantly better suited to function as an attractor network than the permuted null model. The table contains the median number of iterations until convergence for the original and permuted connectomes for different temperature parameters $\beta$ and the p-value derived from a one-sided Wilcoxon signed-rank test (i.e. a non-parametric paired test) comparing the iteration values for each random null instance (1000 pairs) to the iteration number observed with the original matrix and the same random input; with the null hypothesis that the empirical connectome converges in fewer iterations. than the permuted connectome. (**H**) We optimized the noise parameter $\epsilon$ of the stochastic relaxation procedure for eight different $\epsilon$ values over a logarithmic range between $\epsilon = 0.1$ and 1 and contrasted the similarity (Wasserstein distance) between the 122-dimensional distribution of the empirical and the fcANN-generated data against null data generated from a covariance-matched multivariate normal distribution (1000 surrogates). We found that the fcANN reached multistability with $\epsilon > 0.19$ and provided the most accurate reconstruction of the real data with $\epsilon = 0.37$, as compared with its accuracy in retaining the null data, suggesting that the fcANN model is capable of capturing non-Gaussian conditionals in the data. Glass's Delta quantifies the distance from the null mean, expressed in units of null standard deviation.

The online version of this article includes the following figure supplement(s) for figure 2:

**Figure supplement 1.** Parameter sweep of functional connectivity-based attractor network (fcANN) parameters threshold and beta.

**Figure supplement 2.** Eigenstructure and projector tests of the functional connectivity-based attractor network (fcANN).

**Figure supplement 3.** Schematic representation of the functional connectivity-based attractor network (fcANN) projection.

**Figure supplement 4.** Functional connectivity-based attractor network s (fcANNs) initialized with the empirical connectome have better convergence properties than permutation-based null models.

To test research question **Q1**, we matched the eigenvectors of the coupling matrix to the attractor state with which they exhibit the highest correlation. We compared eigenvector-attractor correlations with a null model based on phase-randomized surrogate time-series data (NM1). We found that the eigenvectors of the coupling matrix and the attractor states are significantly more strongly aligned (as measured with Pearson's correlation coefficient) than those in the null model (two-sided empirical permutation test; 1000 permutations; correlations and p-values for the first six eigenvector-attractor pairs are reported in *Figure 2A*), providing evidence that large-scale brain organization approximates a Kanter-Sompolinsky projector network architecture (*Figure 2A*). Eigenvectors with the highest eigenvalues tended to be aligned with the attractor states with the highest fractional occupancy (ratio of time spent on their basins during simulations with stochastic relaxation; see *Figure 2F*). No such pattern was observed in the null model (*Figure 2B*). Further evidence for the functional connectome's close resemblance to a Kanter-Sompolinsky projector network is provided by the orthogonality of the attractor states to each other (*Figure 2C*) and additional analyses reported in *Figure 2—figure supplement 2*.

Next, to support the visualization of further analyses, we constructed a simplified, 2-dimensional visual representation of fcANN dynamics, which we apply throughout the remaining manuscript as a high-level visual summary. This 2-dimensional visualization, referred to as the fcANN projection, is based on the first two principal components (PCs) of the states sampled from the stochastic relaxation procedure (*Figure 2D–F*, *Figure 2—figure supplement 3*). On this simplified visualization, we observed a clear separation of the attractor states (*Figure 2D*), with the two symmetric pairs of attractor states located at the extremes of the first and second PC. To map the attractors' basins on the space spanned by the first two PCs (*Figure 2C*), we obtained the attractor state of each point visited during the stochastic relaxation and fit a multinomial logistic regression model to predict the attractor state from the first two PCs. The resulting model accurately predicted attractor states of arbitrary brain activity patterns, achieving a cross-validated accuracy of 96.5% (two-sided empirical

permutation $p<0.001$; 1000 label permutations within folds). This allows us to visualize attractor basins on this 2-dimensional projection by delineating the decision boundaries obtained from this model (*Figure 2—figure supplement 3* as black lines in *Figure 2G–H*). In the rest of the manuscript, we use this 2-dimensional fcANN projection depicted on (*Figure 2D–H*) as a simplified visual representation of brain dynamics.

Panel D on *Figure 2* uses the fcANN projection to visualize the conventional Hopfield relaxation procedure. It depicts the trajectory of individual activation maps (sampled randomly from the time series data in study 1) until converging to one of the four attractor states. Panel E shows that the system does not converge to an attractor state with the stochastic relaxation procedure. The resulting path is still influenced by the attractor states' 'gravitational pull,' resulting in multistable dynamics that resemble the empirical time series data (example data on panel F).

In study 1, we investigated the convergence process of the fcANN (research question **Q2**) and contrasted it with a null model based on permuted variations of the connectome (while retaining the symmetry of the matrix, NM2). This null model preserves the sparseness and the degree distribution of the connectome but destroys its topological structure (e.g. clusteredness). We found that the topology of the original (unpermuted) functional brain connectome makes it significantly better suited to function as an attractor network than the permuted null model. For instance, with $\beta = 0.04$, the median iteration number for the original and permuted fcANNs to reach convergence was 383 and 3543.5 iterations, respectively (*Figure 2G*, *Figure 2—figure supplement 4*). Similar results were observed, independent of the inverse temperature parameter $\beta$. We set the temperature parameter for the rest of the paper to a value of $\beta = 0.04$, resulting in 4 distinct attractor states. The primary motivation for selecting $\beta = 0.04$ was to reduce the computational burden and the interpretational complexity for further analyses. However, as with increasing temperature attractor states emerge in a nested fashion, we expect that the results of the following analyses would be, although more detailed, qualitatively similar with higher $\beta$ values.

Next, in line with research question **Q3**, we optimized the noise parameter $\epsilon$ of the stochastic relaxation procedure for 8 different $\epsilon$ values over a logarithmic range between $\epsilon = 0.1$ and 1 and contrasted the similarity (Wasserstein distance) between the 122-dimensional distribution of the empirical and the fcANN-generated data against null data generated from a covariance-matched multivariate normal distribution (1000 surrogates). We found that the fcANN reached multistability with $\epsilon > 0.19$ and provided the most accurate reconstruction of the non-Gaussian conditional dependencies in the real data with $\epsilon = 0.37$, as compared to its accuracy in retaining the covariance-matched multivariate Gaussian null data (NM3 *Figure 2H*; Wasserstein distance: 10.2, Glass's Delta (distance from null mean, expressed in units of null standard deviation): –11.63, $p<0.001$ one-sided). Based on this coarse optimization procedure, we set $\epsilon = 0.37$ for all subsequent analyses.

## Reconstruction of resting state brain dynamics

Next, we visualized and qualitatively assessed the neuroscientific relevance of the spatial patterns of the obtained attractor states (**Q4**, *Figure 3A*), and found that they closely resemble previously described large-scale brain systems. The spatial patterns associated with the first pair of attractors (mapped on PC1 on the 2-dimensional projection, horizontal axis, e.g. on *Figure 2D–H*) show a close correspondence to two commonly described complementary brain systems that have been previously found in anatomical, functional, developmental, and evolutionary hierarchies, as well as gene expression, metabolism, and blood flow (see *Sydnor et al., 2021* for a review), and reported under various names, like intrinsic and extrinsic systems (*Golland et al., 2008*), Visual-Sensorimotor-Auditory and Parieto-Temporo-Frontal 'rings' (*Cioli et al., 2014*), 'primary' brain substates (*Chen et al., 2018*), unimodal-to-transmodal principal gradient (*Margulies et al., 2016*; *Huntenburg et al., 2018*), or sensorimotor-association axis (*Sydnor et al., 2021*). A common interpretation of these two patterns is that they represent (i) an 'intrinsic' system for higher-level internal context, commonly referred to as the default mode network (*Raichle et al., 2001*) and (ii) an anti-correlated 'extrinsic' system linked to the immediate sensory environment, showing similarities to the recently described 'action mode network' (*Dosenbach et al., 2025*). The other pair of attractors - spanning an approximately orthogonal axis - resemble patterns commonly associated with perception–action cycles (*Fuster, 2004*), and described as a gradient across sensorimotor modalities (*Huntenburg et al., 2018*), recruiting regions associated with active (e.g. motor cortices) and perceptual inference (e.g. visual areas).

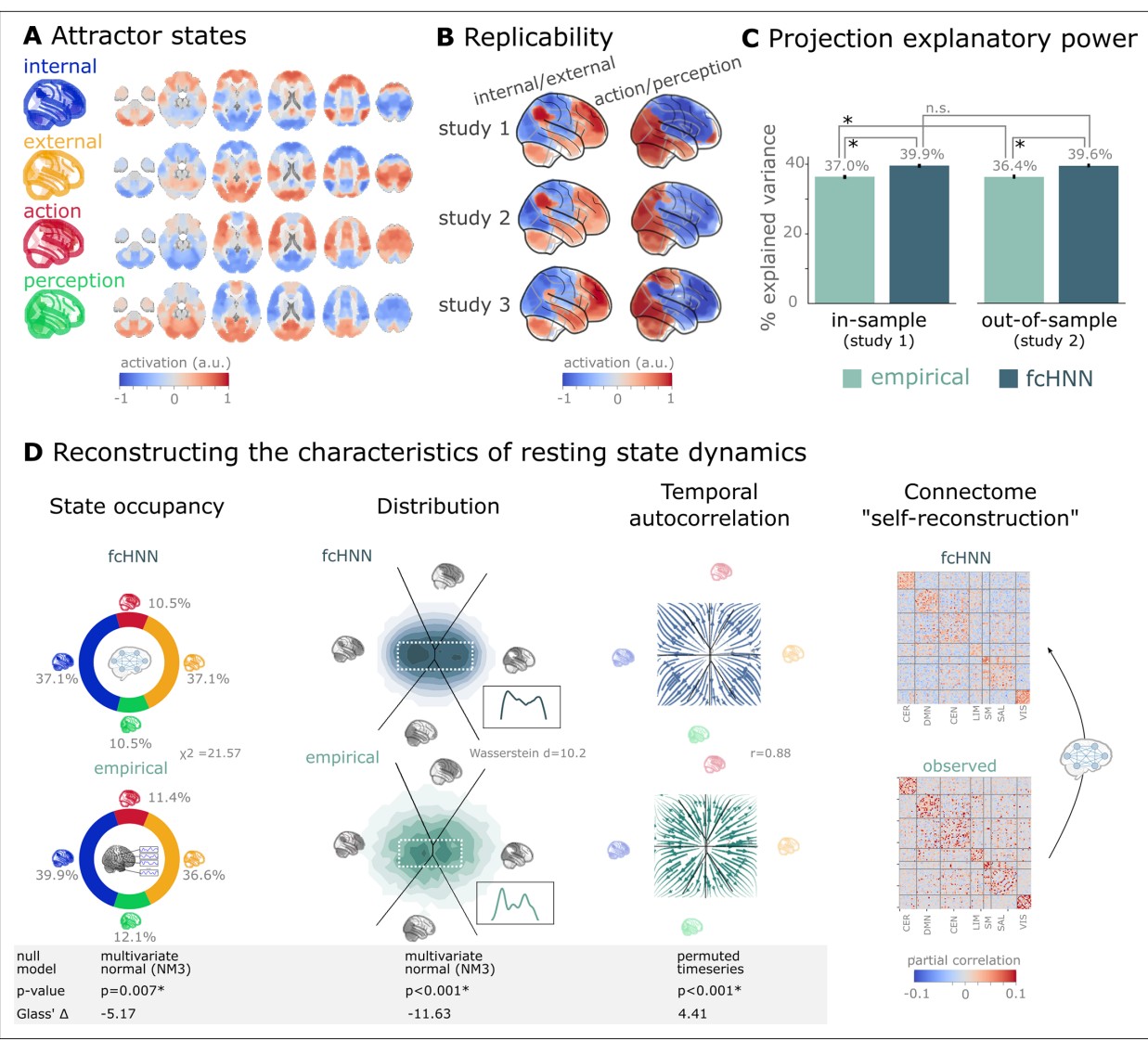

**Figure 3.** Connectome-based attractor networks reconstruct characteristics of real resting-state brain activity. (**A**) The four attractor states of the functional connectivity-based attractor network (fcANN) model from study 1 reflect brain activation patterns with high neuroscientific relevance, representing sub-systems previously associated with 'internal context' (blue), 'external context' (yellow), 'action' (red), and 'perception' (green) (*Golland et al., 2008*; *Cioli et al., 2014*; *Chen et al., 2018*; *Fuster, 2004*; *Margulies et al., 2016*; *Dosenbach et al., 2025*). (**B**) The attractor states show excellent replicability in two external datasets (study 2 and 3, overall mean correlation 0.93). (**C**) The first two principal components (PCs) of the fcANN state space (the 'fcANN projection') explain significantly more variance (two-sided percentile bootstrap $p<0.0001$ on $\Delta R^2$, 100 resamples) in the real resting-state fMRI data than principal components derived from the real resting-state data itself and generalizes better (two-sided percentile bootstrap $p<0.0001$) to out-of-sample data (study 2). Error bars denote 99% percentile bootstrapped confidence intervals (100 resamples). (**D**) The fcANN analysis reliably predicts various characteristics of real resting-state fMRI data, such as the fraction of time spent on the basis of the four attractors (first column, $p=0.007$, contrasted to the multivariate normal null model NM3), the distribution of the data on the fcANN-projection (second column, $p<0.001$, contrasted to the multivariate normal null model NM3) and the temporal autocorrelation structure of the real data (third column, $p<0.001$, contrasted to a null model based on permuting time frames). The latter analysis was based on flow maps of the mean trajectories (i.e. the characteristic timeframe-to-timeframe transition direction) in fcANN-generated data, as compared to a shuffled null model representing zero temporal autocorrelation. For more details, see Methods. Furthermore, we demonstrate that - in line with the theoretical expectations - fcANNs 'leak' their weights during stochastic inference (rightmost column): the time series resulting from the stochastic relaxation procedure mirror the covariance structure of the functional connectome the fcANN model was initialized with. While the 'self-reconstruction' property in itself does not strengthen the face validity of the approach (no unknown information is reconstructed), it is a strong indicator of the model's construct validity; i.e., that systems that behave like the proposed model inevitably 'leak' their weights into the activity time series.

The online version of this article includes the following figure supplement(s) for figure 3:

**Figure supplement 1.** Robustness of the functional connectivity-based attractor network (fcANN) weights to noise.

*Figure 3 continued on next page*

*Figure 3 continued*

**Figure supplement 2.** Statistical inference of the functional connectivity-based attractor network (fcANN) state occupancy prediction with different null models.

**Figure supplement 3.** Explained variance in state energy by first two principal components.

**Figure supplement 4.** Cross-validation classification accuracy of the functional connectivity-based attractor network (fcANN), when predicting the attractor state from state activation.

The discovered attractor states demonstrate high replicability across the discovery dataset (study 1) and two independent replication datasets (study 2 and 3, *Figure 3C*; overall mean Pearson's correlation 0.93, pooled across datasets and attractor states). In a supplementary analysis, we have also demonstrated the robustness of fcANNs to imperfect functional connectivity measures: fcANNs were found to be significantly more robust to noise added to the coupling matrix than nodal strength scores (used as a reference with the same dimensionality; see *Figure 3—figure supplement 1* for details).

Further analysis in study 1 showed that connectome-based attractor models accurately reconstructed multiple characteristics of true resting-state data (**Q5**). First, the two 'axes' of the fcANN projection (corresponding to the first four attractors) accounted for a substantial amount of variance in the real resting-state fMRI data in study 1 (mean $R^2 = 0.399$) and generalized well to out-of-sample data (study 2, mean $R^2 = 0.396$) (*Figure 3E*). The variance explained by the attractors significantly exceeded that of the first two PCs derived directly from the real resting-state fMRI data itself ($R^2 = 0.37$ and 0.364 for in- and out-of-sample analyses). PCA - by identifying variance-heavy orthogonal directions - aims to explain the highest amount of variance possible in the data (with the assumption of Gaussian conditionals). While empirical attractors are closely aligned to the PCs (i.e. eigenvectors of the inverse covariance matrix), the alignment is only approximate. Here, we quantified whether attractor states are a better fit to the unseen data than the PCs. Obviously, due to the otherwise strong PC-attractor correspondence, this is expected to be only a small improvement. However, this provides important evidence for the validity of our framework, as - together with our analysis addressing Q3 - it shows that attractors are not just a complementary, perhaps 'noisier' variety of the PCs, but a 'substrate' that generalizes better to unseen data than the PCs themselves.

Second, during stochastic relaxation, the fcANN model was found to spend approximately three-quarters of the time on the basins of the first two attractor states and one-quarter on the basins of the second pair of attractor states (approximately equally distributed between pairs). We observed similar temporal occupancies in the real data (*Figure 3D* left column), statistically significant against a covariance-matched multivariate Gaussian null model (NM3, 1000 surrogates each; observed $\chi^2 = 21.57$, $p<0.001$; Glass $\Delta = -5.17$; see *Figure 3—figure supplement 2* for details and for an alternative null model based on spatial phase-randomization). Fine-grained details of the distribution with bimodal appearance, observed in the real resting-state fMRI data were also convincingly reproduced by the fcANN model (*Figures 2D and 3F*, second column).

Third, not only spatial activity patterns but also time series generated by the fcANN are similar to empirical time series data. Next to the visual similarity shown on *Figure 2E and F*, we observed a statistically significant similarity between the average trajectories of fcANN-generated and real time series 'flow' (i.e. the characteristic timeframe-to-timeframe transition direction, Pearson's $r=0.88$, $p<0.001$, Glass $\Delta = 4.41$), as compared to null models of zero temporal autocorrelation (randomized timeframe order; two-sided empirical permutation test on Pearson's r with 1000 permutations; *Figure 3D*, third column; Methods).

Finally, we have demonstrated that, as expected from theory, fcANNs generate signal that preserves the covariance structure of the functional connectome they were initialized with, indicating that dynamic systems of this type (including the brain) inevitably 'leak' their underlying structure into the activity time series, strengthening the construct validity of our approach (*Figure 3D*).

## An explanatory framework for task-based brain activity

Next to reproducing various characteristics of spontaneous brain dynamics, fcANNs can also be used to model responses to various perturbations (**Q6**). We obtained task-based fMRI data from a study by *Woo et al., 2015* (study 4, n=33, see *Figure 3*), investigating the neural correlates of pain and its self-regulation.

We found that activity changes due to pain (taking into account hemodynamics, see Methods) were characterized on the fcANN projection by a shift toward the attractor state of action/execution (NM5: two-sided permutation test on the L2 norm of the mean projection difference; 1000 within-participant label swaps; *p*<0.001; Glass's Δ = 4.34; *Figure 4A*, left). Energies, as defined by the fcANN, were also significantly different between the two conditions (NM5: two-sided permutation test on absolute energy difference; 1000 label swaps; *p*<0.001; Glass's Δ = 3.14), with higher energies during pain stimulation.

When participants were instructed to up- or downregulate their pain sensation (resulting in increased and decreased pain reports and differential brain activity in the nucleus accumbens, NAc see *Woo et al., 2015* for details), we observed further changes in the location of momentary brain activity patterns on the fcANN projection (two-sided permutation test on the L2 norm of the mean projection difference; 1000 label swaps; *p*<0.001; Glass's Δ = 4.1; *Figure 4A*, right), with downregulation pulling brain dynamics toward the attractor state of internal context and perception. Interestingly, self-regulation did not trigger significant energy changes (two-sided permutation test on absolute energy difference; 1000 label swaps; *p*=0.37; Glass's Δ = 0.4).

Next, we conducted a 'flow analysis' on the fcANN projection, quantifying how the average timeframe-to-timeframe transition direction differs on the fcANN projection between conditions (see Methods). This analysis unveiled that during pain (*Figure 4B*, left side), brain activity tends to gravitate toward a distinct point on the projection on the boundary of the basins of the internal and action attractors, which we term the 'ghost attractor' of pain (similar to *Vohryzek et al., 2020*). In the case of downregulation (as compared to upregulation), brain activity is pulled away from the pain-related 'ghost attractor' (*Figure 4C*, left side), toward the attractor of internal context.

Our fcANN was able to accurately reconstruct these nonlinear dynamics by adding a small amount of realistic 'control signal' (similarly to network control theory, see e.g. *Liu et al., 2011* and *Gu et al., 2015*). To simulate the alterations in brain dynamics during pain stimulation, we acquired a meta-analytic pain activation map (*Zunhammer et al., 2021*) (n=603) and incorporated it as a control signal added to each iteration of the stochastic relaxation procedure. The ghost attractor found in the empirical data was present across a relatively wide range of signal-to-noise (SNR) values (*Figure 4—figure supplement 1*). Results with SNR = 0.005 are presented in *Figure 4B*, right side (Pearson's *r*=0.46; two-sided permutation *p*=0.005 based on NM5: randomizing conditions on a per-participant basis; 1,000 permutations; Glass's Δ = 2.19).

The same model was also able to reconstruct the observed nonlinear differences in brain dynamics between the up- and downregulation conditions (Pearson's *r*=0.62; *p*=0.023 based on two-sided permutation test NM5: randomly shuffling conditions in a per-participant basis; 1000 permutations; Glass's Δ = 1.84) without any further optimization (SNR = 0.005, *Figure 4C*, right side). The only change we made to the model was the addition (downregulation) or subtraction (upregulation) of control signal in the NAc (the region in which *Woo et al., 2015* observed significant changes between up- and downregulation), introducing a signal difference of ΔSNR = 0.005 (the same value we found optimal in the pain analysis). Results were reproducible with lower NAc SNRs, too (*Figure 4—figure supplement 2*).

To provide a comprehensive picture on how tasks and stimuli other than pain map onto the fcANN projection, we obtained various task-based meta-analytic activation maps from Neurosynth (see Methods) and plotted them on the fcANN projection (*Figure 4E*). This analysis reinforced and extended our interpretation of the four investigated attractor states and shed more light on how various functions are mapped on the axes of internal vs. external context and perception vs. action. In the coordinate system of the fcANN projection, visual processing is labeled 'external-perception,' sensory-motor processes fall into the 'external-active' domain, language, verbal cognition and working memory belong to the 'internal-active' region, and long-term memory, as well as social and autobiographic schemata fall into the 'internal-perception' regime (*Figure 4F*).

## Clinical relevance

To demonstrate fcANN models' potential to capture altered brain dynamics in clinical populations (**Q7**), we obtained data from n=172 autism spectrum disorder (ASD) and typically developing control (TDC) individuals, acquired at the New York University Langone Medical Center, New York, NY, USA (NYU) and generously shared in the Autism Brain Imaging Data Exchange dataset (study 7: ABIDE,

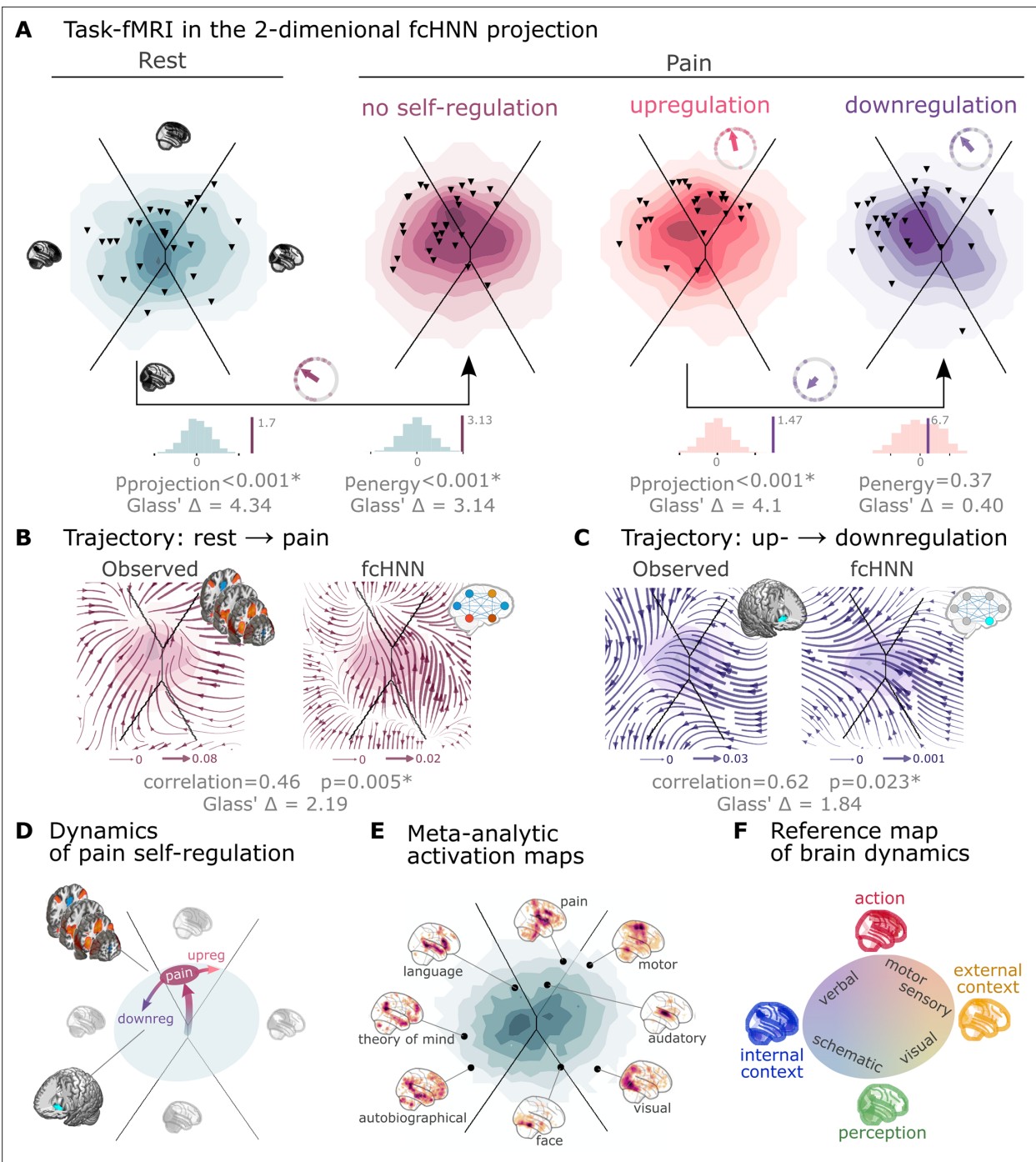

**Figure 4.** Functional connectivity-based attractor networks reconstruct real task-based brain activity. (**A**) Functional MRI time frames during pain stimulation from study 4 (second functional connectivity-based attractor network fcANN projection plot) and self-regulation (third and fourth) are distributed differently on the fcANN projection than brain substates during rest (first projection, permutation test, *p*<0.001 for all). Energies, as defined by the Hopfield model, are also significantly different between rest and the pain conditions (permutation test, *p*<0.001), with higher energies during pain stimulation. Triangles denote participant-level mean activations in the various blocks (corrected for hemodynamics). Small circle plots show the directions of the change for each individual (points) as well as the mean direction across participants (arrow), as compared to the reference state (downregulation for the last circle plot, rest for all other circle plots). (**B**) Flow analysis (difference in the average timeframe-to-timeframe transition direction) reveals a nonlinear difference in brain dynamics during pain and rest (left). When introducing a weak pain-related signal in the fcANN model during stochastic relaxation, it accurately reproduces these nonlinear flow differences (right). (**C**) Simulating activity in the Nucleus Accumbens (NAc) (the region showing significant activity differences in ***Woo et al., 2015***) reconstructs the observed nonlinear flow difference between up- and downregulation (left). (**D**) Schematic representation of brain dynamics during pain and its up- and downregulation, visualized on the fcANN projection.

*Figure 4 continued on next page*

*Figure 4 continued*

In the proposed framework, pain does not simply elicit a direct response in certain regions, but instead, shifts spontaneous brain dynamics towards the 'action' attractor, converging to a characteristic 'ghost attractor' of pain. Down-regulation by NAc activation exerts force towards the attractor of internal context, leading to the brain less frequent 'visiting' pain-associated states. (E) Visualizing meta-analytic activation maps (see *Figure 4—source data 1* for details) on the fcANN projection captures intimate relations between the corresponding tasks and (F) serves as a basis for a fcANN-based theoretical interpretative framework for spontaneous and task-based brain dynamics. In the proposed framework, task-based activity is not a mere response to external stimuli in certain brain locations but a perturbation of the brain's characteristic dynamic trajectories, constrained by the underlying functional connectivity. From this perspective, 'activity maps' from conventional task-based fMRI analyses capture time-averaged differences in these whole brain dynamics.

The online version of this article includes the following source data and figure supplement(s) for figure 4:

**Source data 1.** Source data detailing the search terms used, and the number of studies included in the meta-analysis, as well as the total number of reported activations, and the maximal Z-statistic.

**Figure supplement 1.** Functional connectivity-based attractor network (FcANN) can reconstruct the pain 'ghost attractor.'.

**Figure supplement 2.** Functional connectivity-based attractor network (fcANN) can reconstruct the changes in brain dynamics caused by the voluntary downregulation of pain (as contrasted to upregulation).

---

*di Martino et al., 2014*). After excluding high-motion cases (with the same approach as in studies 1–4, see Methods), we visualized the distribution of time-frames on the fcANN-projection separately for the ASD and TDC groups (*Figure 5A*). First, we assigned all timeframes to one of the 4 attractor states with the fcANN from study 1 and found several significant differences in the mean activity on the attractor basins (see Methods) of the ASD group as compared to the respective controls (*Figure 5B*). Strongest differences were found on the 'action-perception' axis (*Table 3*), with increased activity of the sensory-motor and middle cingular cortices during 'action-execution' related states and increased visual and decreased sensory and auditory activity during 'perception' states, likely reflecting the widely acknowledged, yet poorly understood, perceptual atypicalities in ASD (*Hadad and Schwartz, 2019*). ASD-related changes in the internal-external axis were characterized by more involvement of the posterior cingulate, the precuneus, the nucleus accumbens, the dorsolateral prefrontal cortex (dlPFC), the cerebellum (Crus II, lobule VII) and inferior temporal regions during activity of the internalizing subsystem (*Table 3*). While similar, default mode network (DMN)-related changes have often been attributed to an atypical integration of information about the 'self' and the 'other' (*Padmanabhan et al., 2017*), a more detailed fcANN analysis may help to further disentangle the specific nature of these changes.

Thus, we contrasted the characteristic trajectories derived from the fcANN models of the two groups (initialized with the group-level functional connectomes). Our fcANN-based flow analysis predicted that in ASD, there is an increased likelihood of states returning towards the middle (more noisy states) from the internal-external axis and an increased likelihood of states transitioning towards the extremes of the action-perception axis (*Figure 5C*). We observed a highly similar pattern in the real data (Pearson's r=0.66), statistically significant after two-sided permutation testing (shuffling the group assignment; 1000 permutations; p=0.009).

## Discussion

The notion that the brain functions as an attractor network has long been proposed (*Freeman, 1987*; *Amit, 1989*; *Deco and Jirsa, 2012*; *Deco et al., 2012*; *Golos et al., 2015*; *Hansen et al., 2015*; *Vohryzek et al., 2020*), although the exact functional form of the network underlying large-scale brain dynamics, or at least useful approximation thereof, remained elusive. The theoretical framework of free energy minimizing attractor neural networks (FEP-ANN) (*Spisak and Friston, 2025*) identifies a specific class of attractor networks that emerge from first principles of self-organization, as articulated by the FEP (*Friston, 2010*; *Friston et al., 2023*). Therefore, it provides a plausible candidate model for large-scale brain attractor dynamics and yields testable predictions - measurable signatures of these special attractor networks that can be validated empirically. In this study, we have introduced and performed initial validation of a simple and robust network-level generative computational model, rooted in the FEP-ANN framework and providing the opportunity to test these predictions empirically. Our model, termed a functional connectivity-based attractor network (fcANN), exploits special characteristics of the emergent inference rule of FEP-ANNs. This is a stochastic rule that governs

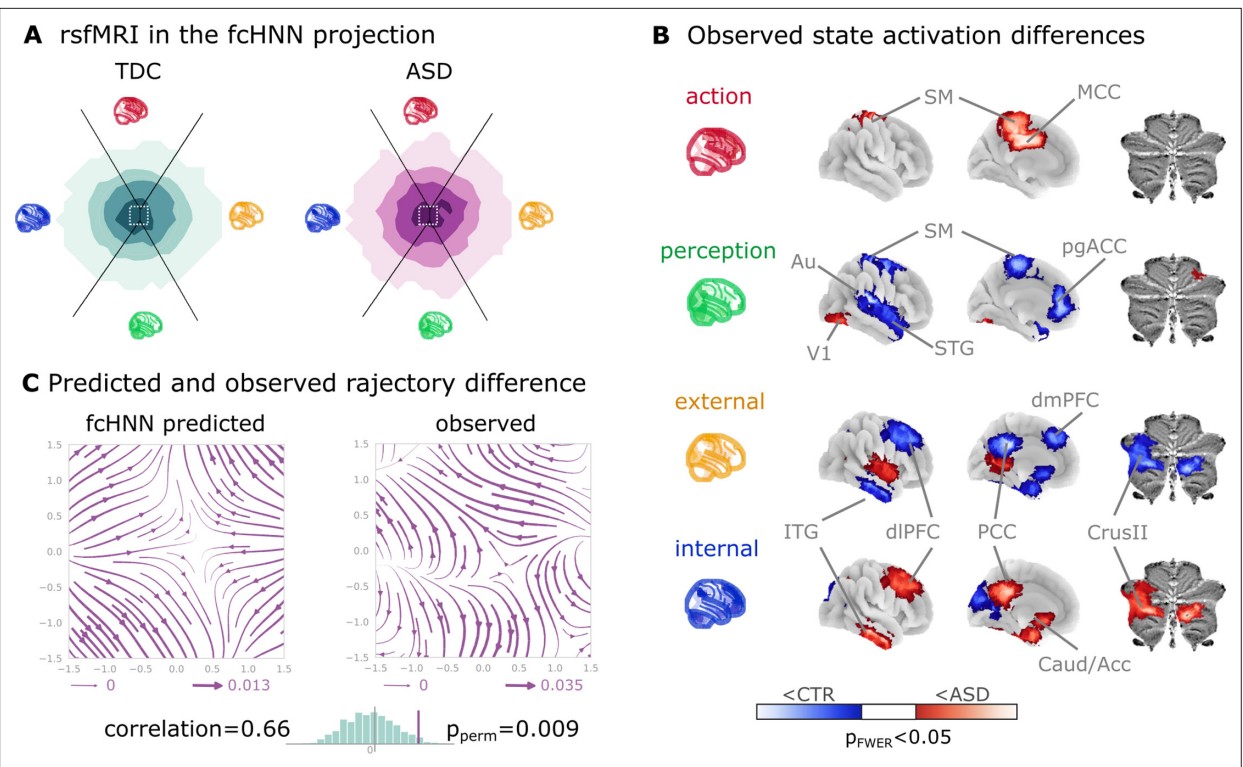

**Figure 5.** Connectome-based Hopfield analysis of autism spectrum disorder. (**A**) The distribution of time frames on the functional connectivity-based attractor network (fcANN) projection separately for ASD patients and typically developing control (TDC) participants. (**B**) We quantified attractor state activations in the Autism Brain Imaging Data Exchange datasets (study 7) as the individual-level mean activation of all time frames belonging to the same attractor state. This analysis captured alterations similar to those previously associated with ASD-related perceptual atypicalities (visual, auditory, and somatosensory cortices) as well as atypical integration of information about the 'self' and the 'other' (default mode network regions). All results are corrected for multiple comparisons across brain regions and attractor states (122×4 comparisons) with Bonferroni correction. See ***Table 3*** and ***Figure 5—source data 1*** for detailed results. (**C**) The comparison of data generated by fcANNs initialized with ASD and TDC connectomes, respectively, revealed a characteristic pattern of differences in the system's dynamics, with increased pull towards (and potentially a higher separation between) the action and perception attractors and a lower tendency of trajectories going towards the internal and external attractors. **Abbreviations**: MCC: middle cingulate cortex, ACC: anterior cingulate cortex, pg: perigenual, PFC: prefrontal cortex, dm: dorsomedial, dl: dorsolateral, STG: superior temporal gyrus, ITG: inferior temporal gyrus, Caud/Acc: caudate-accumbens, SM: sensorimotor, V1: primary visual, A1: primary auditory, SMA: supplementary motor cortex, ASD: autism spectrum disorder, TDC: typically developing control.

The online version of this article includes the following source data for figure 5:

**Source data 1.** All significant differences of the mean state activation analysis on the ABIDE dataset; label denotes the region in the BASC122 atlas.

how activity evolves in time with a given set of fixed coupling weights and leads to a Markov chain Monte Carlo (MCMC) sampling process. As a consequence, the activation time series data measured in each network node can be used to reconstruct the attractor network's internal structure. Specifically, the coupling weights can be estimated as the negative inverse covariance matrix of the activation time series data. fcANN modeling applies this concept to large-scale brain dynamics as measured by resting-state fMRI data (as an estimate of weights corresponding to the steady-state distribution of the system).

The core idea underlying the fcANN reconstruction approach - the use of functional connectivity as a proxy for weighted information flow in the brain - is in line with previous empirical observations about the relationship between functional connectivity and brain activity, as articulated by the activity flow principle, first introduced by ***Cole et al., 2016***. The activity flow principle states that activity in a brain region can be predicted by a weighted combination of the activity of all other regions, where the weights are set to the functional connectivity of those regions to the held-out region. This principle has been shown to hold across a wide range of experimental and clinical conditions (***Cole et al., 2016***; ***Ito et al., 2017***; ***Mill et al., 2022***; ***Hearne et al., 2021***; ***Chen et al., 2018***). Considering that the repeated, iterative application of the activity flow equation (extended with an arbitrary sigmoidal

**Table 3.** The top ten largest changes in average attractor-state activity between autistic and control individuals.
Mean attractor-state activity changes are presented in the order of their absolute effect size. Reported effect sizes are mean attractor activation differences. Note that activation time series were standard scaled independently for each region, so effect size can be interpreted as showing the differences as a proportion of regional variability. All p-values are based on permutation tests (shuffling the group assignment) and corrected for multiple comparisons (via Bonferroni's correction). For a comprehensive list of significant findings, see *Figure 5—source data 1*.

| Region | Attractor | Effect size | p-value |
|---|---|---|---|
| Primary auditory cortex | Perception | –0.126 | <0.0001 |
| Middle cingulate cortex | Action | 0.109 | <0.0001 |
| Cerebellum lobule VIIb (medial part) | Internal context | 0.104 | <0.0001 |
| Mediolateral sensorimotor cortex | Perception | –0.099 | 0.00976 |
| Precuneus | Action | 0.098 | <0.0001 |
| Middle superior temporal gyrus | Perception | –0.098 | <0.0001 |
| Frontal eye field | Perception | –0.095 | <0.0001 |
| Dorsolateral sensorimotor cortex | Perception | –0.094 | 0.00976 |
| Posterior cingulate cortex | Action | 0.092 | <0.0001 |
| Dorsolateral prefrontal cortex | External context | –0.092 | <0.0001 |

activation function) naturally reproduces certain types of recurrent artificial neural networks, e.g., Hopfield networks (*Hopfield, 1982*) yield an intuitive understanding of how the fcANN model works.

However, beyond this analogy, we need concrete evidence that the fcANN model and the underlying FEP-ANN framework is a valid model of large-scale brain dynamics.

Here, we have tested multiple predictions of the FEP-ANN framework. Most importantly, FEP-ANNs - through their emergent predictive coding-based learning rule - have been shown to develop approximately orthogonal attractor representations, and thereby approximate the so-called Kanter-Sompolinsky projector neural networks (K-S network, for short) (*Kanter and Sompolinsky, 1987*). K-S networks are a special class of attractor networks that have been shown to be highly effective for pattern recognition and learning (*Kanter and Sompolinsky, 1987*). In these networks, the attractor states are orthogonal to each other and become equivalent with those eigenvectors of the coupling matrix that have positive eigenvalues. This is a very strong prediction: K-S networks are a very special class of attractor networks, which do not arise from conventional learning rules, like Hebbian learning. To date, the predictive coding-based learning rule of FEP-ANNs is the only known local, incremental learning rule that can effectively approximate K-S networks in a single phase (but see 'dreaming neural networks,' *Fachechi et al., 2019* for a similar, two-phase approach). Thus, showing that fcANN models approximate K-S networks can be interpreted as evidence for the brain functioning akin to a FEP-ANN. Our results show that this is indeed the case: the fcANN models reconstructed from resting-state fMRI data approximate K-S networks, and thereby exhibit approximately orthogonal attractor states.

In the FEP-ANN framework, approximate attractor orthogonality has important computational implications: it allows the system to achieve maximal 'storage capacity' (the number of attractors that can be stored and retrieved without interference). Furthermore, in the FEP-ANN framework, attractor states can be interpreted as learned priors that capture the statistical structure of the environment, while the stochastic dynamics implement posterior sampling. Orthogonal attractors emerge as an efficient way to span the subspace of interest, fostering generalization to unseen data (as long as it is from the subspace spanned by the existing attractors).

Next, we have demonstrated that fcANN models exhibit multiple biologically plausible attractor states, with large basins, and showed that the relaxation dynamics of the reconstructed model display fast convergence to attractor states - another signature that the functional connectome being a valid proxy for the coupling weights of an underlying attractor network. Relying on previous work, we can establish a relatively straightforward (although somewhat speculative) correspondence between attractor states and brain function, mapping brain activation on the axes of internal vs. external

context (*Golland et al., 2008*; *Cioli et al., 2014*), as well as perception vs. action (*Fuster, 2004*). In our framework, the attractor states can be interpreted as learned priors that capture the statistical structure of the environment, while the stochastic dynamics implement posterior sampling. This connection suggests that canonical resting-state networks may represent the brain's internal generative model of the world, continuously updated through the emergent learning dynamics we described theoretically. Furthermore, the relation between fcANN models and the FEP-ANN framework substantiates that the reconstructed attractor states are not solely local minima in the state space but act as a driving force for the dynamic trajectories of brain activity. We argue that attractor dynamics may be the main driving factor for the spatial and temporal autocorrelation structure of the brain, recently described to be predictive of network topology in relation to age, subclinical symptoms of dementia, and pharmacological manipulations with serotonergic drugs (*Shinn et al., 2023*). Nevertheless, attractor states should not be confused with the conventional notion of brain substates (*Chen et al., 2015*; *Kringelbach and Deco, 2020*) and large-scale functional gradients (*Margulies et al., 2016*). In the fcANN framework, attractor states can rather be conceptualized as 'Platonic idealizations' of brain activity, that are continuously approximated - but never reached - by the brain, resulting in re-occurring patterns (brain substates) and smooth gradual transitions (large-scale gradients).

Considering the functional connectome as weights of a neural network distinguishes our methodology from conventional biophysical and phenomenological computational modeling strategies, which usually rely on the structural connectome to model polysynaptic connectivity (*Cabral et al., 2017*; *Deco et al., 2012*; *Golos et al., 2015*; *Hansen et al., 2015*). Given the challenges of accurately modeling the structure-function coupling in the brain (*Seguin et al., 2023*), such models are currently limited in terms of reconstruction accuracy, hindering translational applications. By working with direct, functional MRI-based activity flow estimates, fcANNs bypass the challenge of modeling the structural-functional coupling and are able to provide a more accurate representation of the brain's dynamic activity propagation (although at the cost of losing the ability to provide biophysical details on the underlying mechanisms). Another advantage of the proposed model is its simplicity. While many conventional computational models rely on the optimization of a high number of free parameters, the basic form of the fcANN approach comprises solely two, easily interpretable 'hyperparameters' (temperature and noise) and yields notably consistent outcomes across an extensive range of these parameters (*Figure 3—figure supplement 3*, *Figure 2—figure supplement 1*, *Figure 3—figure supplement 2*, *Figure 4—figure supplement 1*, *Figure 4—figure supplement 2*). To underscore the potency of this simplicity and stability, in the present work, we avoided any unnecessary parameter optimization, leaving a negligible chance of overfitting. It is likely, however, that extensive parameter optimization could further improve the accuracy of the model.

Furthermore, the fcANN approach links brain dynamics directly to dynamical systems theory and the free energy principle, conceptualizes the emergence of large-scale canonical brain networks (*Zalesky et al., 2014*) in terms of multistability, and sheds light on the origin of characteristic task responses that are accounted for by 'ghost attractors' in the system (*Deco and Jirsa, 2012*; *Vohryzek et al., 2020*). As fcANNs do not need to be trained to solve any explicit tasks, they are well-suited to examine spontaneous brain dynamics. However, it is worth mentioning that fcANNs can also be further trained via the predictive coding-based learning rule of FEP-ANNs, to 'solve' various tasks or to match developmental dynamics or pathological alterations. In this promising future direction, the training procedure itself becomes part of the model, providing testable hypotheses about the formation and various malformations of brain dynamics. A promising application of this is to consider structural brain connectivity (as measured by diffusion MRI) as a sparsity constraint for the coupling weights and then train the fcANN model to match the observed resting-state brain dynamics. If the resulting structural-functional ANN model is able to closely match the observed functional brain substate dynamics, it can be used as a novel approach to quantify and understand the structural-functional coupling in the brain.

Given its simplicity, it is noteworthy how well the fcANN model is able to reconstruct and predict brain dynamics under a wide range of conditions. First and foremost, we have found that the topology of the functional connectome seems to be well-suited to function as an attractor network, as it converges much faster than the respective null models. Second, we found that the two-dimensional fcANN projection can explain more variance in real (unseen) resting-state fMRI data than the first two principal components derived from the data itself. This may indicate that through the known noise

tolerance of attractor neural networks, fcANNs are able to capture essential characteristics of the underlying dynamic processes even if our empirical measurements are corrupted by noise and low sampling rate. Indeed, fcANN attractor states were found to be robust to noisy weights (*Figure 3—figure supplement 1*) and highly replicable across datasets acquired at different sites, with different scanners and imaging sequences (studies 2 and 3). The observed high level of replicability allowed us to reuse the fcANN model constructed with the functional connectome of study 1 for all subsequent analyses, without any further fine-tuning or study-specific parameter optimization.

Both conceptually and in terms of analysis practices, resting and task states are often treated as separate phenomena. However, in the fcANN framework, the differentiation between task and resting states is considered an artificial dichotomy. Task-based brain activity in the fcANN framework is not a mere response to external stimuli in certain brain locations but a perturbation of the brain's characteristic dynamic trajectories, with increased preference for certain locations on the energy landscape ('ghost attractors'). In our analyses, the fcANN approach captured and predicted participant-level activity changes induced by pain and its self-regulation and gave a mechanistic account for how relatively small activity changes in a single region (NAcc) may result in a significantly altered pain experience. Our control-signal analysis is different from, but compatible with, linear network control theory-based approaches (*Liu et al., 2011*; *Gu et al., 2015*). Combining network control theory with the fcANN approach could provide a powerful framework for understanding the effects of various tasks, conditions, and interventions (e.g. brain stimulation) on brain dynamics.

Brain dynamics can not only be perturbed by task or other types of experimental or naturalistic interventions, but also by pathological alterations. Here, we provide an initial demonstration (study 7) of how fcANN-based analyses can characterize and predict altered brain dynamics in autism spectrum disorder (ASD). The observed ASD-associated changes in brain dynamics are indicative of a reduced ability to flexibly switch between perception and internal representations, corroborating previous findings that in ASD, sensory-driven connectivity transitions do not converge to transmodal areas (*Hong et al., 2019*). Such findings are in line with previous reports of a reduced influence of context on the interpretation of incoming sensory information in ASD (e.g. the violation of Weber's law) (*Hadad and Schwartz, 2019*).

Our findings open up a series of exciting opportunities for the better understanding of brain function in health and disease. First, fcANN analyses may provide insights into the causes of changes in brain dynamics by, for instance, identifying the regions or connections that act as an 'Achilles' heel' in generating such changes. Such control analyses could, for instance, aid the differentiation of primary causes and secondary effects of activity or connectivity changes in various clinical conditions. Rather than viewing pathology as static connectivity differences, our approach suggests that disorders may reflect altered attractor landscapes that bias brain dynamics toward maladaptive states. This perspective could inform the development of targeted interventions that aim to reshape these landscapes through neurofeedback, brain stimulation, or pharmacological approaches.

Second, as a generative model, fcANNs provide testable predictions about the effects of various interventions on brain dynamics, including pharmacological modulations as well as non-invasive brain stimulation (e.g. transcranial magnetic or direct current stimulation, focused ultrasound, etc.) and neurofeedback. Obtaining the optimal stimulation or treatment target within the fcANN framework (e.g. by means of network control theory, *Liu et al., 2011*) is one of the most promising future directions with the potential to significantly advance the development of novel, personalized treatment approaches.

Third, the theoretical integration of the fcANN model with the FEP-ANN framework positions our work within a broader scientific program that seeks to understand the brain as a self-organizing, information-processing system governed by fundamental physical and computational principles. The empirical validation of attractor orthogonality represents a crucial step toward establishing this unified framework for understanding brain function across scales and contexts.

The proposed approach is not without limitations. First, fcANNs do not incorporate information about anatomical connectivity and do not explicitly model biophysical details. Thus, in its present form, the model is not suitable to study the structure-function coupling and cannot yield mechanistic explanations underlying (altered) polysynaptic connections, at the level of biophysical details. Nevertheless, our approach showcases that many characteristics of brain dynamics, like multistability, temporal autocorrelations, states, and gradients, can be explained and predicted by a very simple

nonlinear phenomenological model. Second, our model assumes a stationary functional connectome, which seems to contradict notions of dynamic connectivity. However, while the underlying FEP-ANN framework focuses on the long-term steady-state distribution of the system, it also naturally incorporates multistable fluctuations and the related dynamic connectivity changes through the stochastic relaxation dynamics. This is in line with the notion of 'latent functional connectivity,' an intrinsic brain network architecture built up from connectivity properties that are persistent across brain substates (*McCormick et al., 2022*).

In this initial work, we presented the simplest possible implementation of the fcANN concept. It is clear that the presented analyses exploit only a small proportion of the richness of the full state-space dynamics reconstructed by the fcANN model. There are many potential ways to further improve the utility of the fcANN approach. Increasing the number of reconstructed attractor states (by increasing the temperature parameter), investigating higher-dimensional dynamics, fine-tuning the hyperparameters, and testing the effect of different initializations and perturbations are all important directions for future work, with the potential to further improve the model's accuracy and usefulness.

## Conclusion

Here, we have proposed a principled, lightweight, theory-driven framework that instantiates the inference dynamics of free-energy-minimizing attractor networks (FEP-ANNs) to model large-scale brain activity. Initialized with empirical functional connectivity, the fcANN links brain connectivity to activity and identifies neurobiologically meaningful attractor states underlying large-scale brain dynamics. We demonstrated that the fcANN models display signs of attractor self-orthogonalization, a hallmark of FEP-ANN systems. The proposed framework provides a simple, interpretable, and predictive basis for studying rest, task perturbations, and disease, and for model-guided interventions.

## Methods
### Statistical analysis and reporting

Unless stated otherwise, all hypothesis tests are two-sided. When permutation tests are used, reported p-values are empirical permutation p-values computed as the proportion of null replicates with a test statistic at least as extreme as the observed one (with the usual +1 numerator/denominator correction). We report the statistic used in each comparison (e.g. Pearson correlation r, chi-square dissimilarity, L2 norm of position differences, median iterations to convergence). Degrees of freedom are reported where parametric tests are used; permutation-based p-values do not have degrees of freedom. By default, permutation counts were 1000, except where explicitly noted (e.g. 50,000 label shuffles in clinical analyses). Confidence intervals based on bootstrap are percentile bootstrap intervals; where shown as 99% CIs, they reflect the percentile range from the bootstrap samples described below.

For multiple comparisons in the clinical analyses, we applied Bonferroni correction across regions and attractor states as specified below.

**Table 4.** Datasets and studies.
The table includes details about the study modality, analysis aims, sample size used for analyses, mean age, gender ratio, and references.

| Study | Modality | Analysis | n | Age (mean ±sd) | % Female | References |
|---|---|---|---|---|---|---|
| Study 1 | Resting state | Discovery | 41 | 26.1±3.9 | 37% | *Spisak et al., 2020* |
| Study 2 | Resting state | Replication | 48 | 24.9±3.5 | 54% | *Spisak et al., 2020* |
| Study 3 | Resting state | Replication | 29 | 24.8±3.1 | 53% | *Spisak et al., 2020* |
| Study 4 | Task-based | Pain self-regulation | 33 | 27.9±9.0 | 66% | *Woo et al., 2015* |
| Study 5 (Meta-analysis) | Task-based | IPD meta-analysis pain map | n=603 (20 studies) | 26.3±5.9 | 39% | *Zunhammer et al., 2021* |
| Study 6 (Neurosynth) | Task-based | Coordinate-based meta-analyses | 14,371 studies in total | N/A | N/A | *Yarkoni et al., 2011* |
| Study 7 (ABIDE, NYU sample) | Resting state | Autism Spectrum Disorder | ASD: 98; NC: 74 | 15.3±6.6 | 20.9% | *di Martino et al., 2014* |

**Table 5.** MRI acquisition parameters.

TR: repetition time; TE: echo time; FA: flip angle; FOV: field of view; EPI: echo-planar imaging; SPGR: spoiled gradient recall; SENSE/GRAPPA/ASSET: parallel imaging factors. Studies 5–7 are meta-analyses or multi-center studies with varying data. Sequence parameters for these studies are available in the respective publications.

| Parameter | Study 1 | Study 2 | Study 3 | Study 4 |
|---|---|---|---|---|
| Scanner/head coil | Philips Achieva X 3T; 32-ch | Siemens Magnetom Skyra 3T; 32-ch | GE Discovery MR750w 3T; 20-ch | Philips Achieva TX 3T; head coil per site |
| Anatomical sequence | T1 MPRAGE | T1 MPRAGE | T1 3D IR-FSPGR | T1 SPGR (high-resolution) |
| Anatomical TR/TE | 8500 ms/3.9 ms | 2300 ms/2.07 ms | 5.3 ms/2.1 ms | -/- |
| Anatomical resolution/FOV | $1\times1\times1$ mm³; $256\times256\times220$ mm³ | $1\times1\times1$ mm³; $256\times256\times192$ mm³ | $1\times1\times1$ mm³; $256\times256\times172$ | - |
| Resting-state EPI TR/TE/FA | 2500 ms/35 ms/90° | 2520 ms/35 ms/90° | 2500 ms/27 ms/81° | 2000 ms/20 ms/ - |
| Phase enc. | COL | A>>P | A>>P | - |
| FOV (voxels×slices) | $240\times240\times132$; 40 slices | $230\times230\times132$; 38 slices | $96\times96\times44$; 44 slices | $64\times64$; 42 slices |
| Slice thickness/gap/order | 3 mm/0.3 mm/interleaved | 3 mm/0.48 mm/interleaved | 3 mm/0 mm/interleaved | 3 mm / - / interleaved |
| Acceleration/fat suppression | SENSE 3×/SPIR | GRAPPA 2×/Fat sat. | ASSET 2×/Fat sat. | SENSE 1.5×/- |
| Volumes/dummies/scan time | 200/5 / 8 min 37 s | 290/5 / 12 min 11 s | 240/0 / 10 min | -/-/- |

## Data

We obtained functional MRI data from seven sources (see *Table 4*). MRI sequence parameters for studies 1–4 are summarized in *Table 5*. We included three resting-state studies with healthy volunteers (study 1, study 2, study 3, $n_{total} = 118$), one task-based study (study 4, $n_{total} = 33$ participants, nine runs each), an individual-participant meta-analytic activation map of pain (study 5, $n_{total} = 603$ from 20 studies), eight task-based activation patterns obtained from coordinate-based meta-analyses via Neurosynth (study 6, 14,371 studies in total; see *Figure 4—source data 1*), and a resting-state dataset focusing on autism spectrum disorder (ASD) from ABIDE (study 7, $n_{total} = 1,112$, *di Martino et al., 2014*).

Study 1 was used to evaluate whether the resting-state functional connectome can be treated as an attractor network, to optimize the temperature ($\beta$) and noise ($\epsilon$) parameters of the fcANN model, and to evaluate the proposed approach for reconstructing resting-state brain dynamics. Studies 2 and 3 served as replication datasets. Studies 1–3 are well suited to examine replicability and generalizability; data were acquired in three centers across two countries, by different research staff, with different scanners (Philips, Siemens, GE) and imaging sequences. Further details on studies 1–3 are described in *Spisak et al., 2020*. The ability of the proposed approach to model task-based perturbations of brain dynamics was evaluated in study 4, which consisted of nine task-based fMRI runs for each of the 33 healthy volunteers. In all runs, participants received heat pain stimulation. Each stimulus lasted 12.5 s, with a 3 s ramp-up and 2 s ramp-down and 7.5 s at target temperature. Six temperature levels were administered (44.3°C, 45.3°C, 46.3°C, 47.3°C, 48.3°C, 49.3°C). We used run 1 (passive experience), run 3 (down-regulation), and run 7 (up-regulation). In runs 3 and 7, participants were asked to cognitively increase (regulate up) or decrease (regulate down) pain intensity. No self-regulation instructions were provided in run 1. See *Woo et al., 2015* for details. Pain control signal for our task-based trajectory analyses on data from study 4 was derived from our individual participant meta-analysis of 20 pain fMRI studies (study 5, n=603). For details, see *Zunhammer et al., 2021*. To obtain fMRI activation maps for other tasks, we used Neurosynth (*Tor, 2011*), a web-based platform for large-scale, automated synthesis of fMRI data. We performed eight coordinate-based meta-analyses with the terms 'motor,' 'auditory,' 'visual,' 'face,' 'autobiographical,' 'theory mind,' 'language,' and 'pain' (*Figure 4—source data 1*) and obtained Z-score maps from a two-way ANOVA, comparing coordinates reported for studies with and without the term of interest and testing for a nonzero association between term use and voxel activation. In study 7 (ABIDE), we obtained preprocessed regional time-series data from the Preprocessed Connectome Project (*Craddock et al., 2013*), as shared at https://

osf.io/hc4md by *Dadi et al., 2019*. Preprocessed time-series data were obtained with the 122-region version of the BASC (Bootstrap Analysis of Stable Clusters) brain atlas (*Bellec et al., 2010*).

## Preprocessing and time-series extraction

Functional MRI data from studies 1–4 were preprocessed with our in-house analysis pipeline, the RPN-pipeline (https://github.com/spisakt/RPN-signature; *Spisak et al., 2022*). The RPN pipeline is based on PUMI (Neuroimaging Pipelines Using Modular workflow Integration, https://github.com/pni-lab/PUMI, *Hoffschlag et al., 2025*) a nipype-based (*Gorgolewski et al., 2011*) workflow management system. It capitalizes on tools from FSL (*Jenkinson et al., 2012*), ANTs (*Avants et al., 2011*), and AFNI (*Cox, 1996*), with code partially adapted from C-PAC (*Craddock et al., 2013*) and niworkflows (*Esteban et al., 2019*), as well as in-house Python routines.

Brain extraction from both anatomical and structural images, as well as tissue segmentation from the anatomical images, was performed with FSL BET and FAST. Anatomical images were linearly and nonlinearly co-registered to the 1 mm MNI152 standard brain template with ANTs (see https://gist.github.com/spisakt/0caa7ec4bc18d3ed736d3a4e49da7415 for parameters). Functional images were co-registered to the anatomical images with FSL FLIRT's boundary-based registration. All resulting transformations were saved for further use. Preprocessing of functional images was performed in native space, without resampling. Realignment-based motion correction was performed with FSL MCFLIRT. The resulting six head motion estimates (three rotations, three translations), their squared terms, their derivatives, and the squared derivatives (Friston-24 expansion, *Friston et al., 1996*) were calculated as nuisance signals. Additionally, head motion was summarized as framewise displacement (FD) time series, according to Power's method (*Power et al., 2012*), for use in censoring and exclusion. After motion correction, outliers (e.g. motion spikes) in time-series data were attenuated using AFNI despike. The union of eroded white-matter maps and ventricle masks was transformed to native functional space and used to extract noise signals for anatomical CompCor correction (*Behzadi et al., 2007*).

In a nuisance regression step, six CompCor parameters (the first six principal components of noise-region time series), the Friston-24 motion parameters, and the linear trend were removed with a general linear model. On the residual data, temporal band-pass filtering was performed with AFNI's 3dBandpass to retain the 0.008–0.08 Hz band. To further attenuate motion artifacts, potentially motion-contaminated time frames, defined by a conservative FD >0.15 mm threshold, were dropped (scrubbing, *Satterthwaite et al., 2013*). Participants were excluded if more than 50% of frames were scrubbed.

The 122-parcel BASC (Bootstrap Analysis of Stable Clusters) atlas (*Bellec et al., 2010*) was individualized by transforming it to each participant's native functional space (nearest-neighbour interpolation) and masking by that participant's gray-matter mask to retain only gray-matter voxels. Voxel time series were then averaged within BASC regions.

All these preprocessing steps are part of the containerized version of the RPN-pipeline (https://spisakt.github.io/RPN-signature), which we run with default parameters for all studies, as in *Spisak et al., 2020*.

## Functional connectome

Regional time series were ordered into large-scale functional modules (defined by the 7-parcel level of the BASC atlas) for visualization. Next, in all datasets, we estimated study-level mean connectivity matrices by regularized partial correlation via the Ledoit–Wolf shrinkage estimator (*Ledoit and Wolf, 2004*), as implemented in Nilearn (*Abraham et al., 2014*). The precision matrix was obtained by inverting the shrinkage covariance, and partial correlations were derived from the precision; diagonal elements were set to zero.

## fcANN inference (FEP-ANN) and update rules

Our fcANN instantiates the inference dynamics of free-energy-minimizing attractor neural networks (FEP-ANNs) at the macro-scale. Each node represents a brain region with continuous activity $\boldsymbol{\sigma} = (\sigma_1, \ldots, \sigma_m)$, and couplings are given by the symmetrized matrix $\mathbf{J}$ (see Functional connectome). Unless noted otherwise, biases are zero ($\boldsymbol{b} = 0$).

Deterministic inference. In the noise-free symmetric case, activities are updated by repeatedly applying a sigmoidal nonlinearity to the weighted input

$$\boldsymbol{\sigma}^{(t+1)} = S\left(\beta J \boldsymbol{\sigma}^{(t)}\right),$$ (5)

where $S$ is a smooth odd sigmoid (we used *tanh* as a practical, fast surrogate for the Langevin function) and $\beta$ is the inverse temperature (precision) scaling the couplings. As the inference rule was derived as a gradient descent on free energy, iterations monotonically decrease the free energy function and, therefore, converge to a local free-energy minimum without any external optimizer. Thus, convergence does not require any optimization procedure with an external optimizer. Instead, it arises as the fixed point of repeated local inference updates, which implement gradient descent on free energy in the deterministic symmetric case (see main text).

Stochastic (Langevin-style) inference. For generative modeling of dynamics, we adopt a slight variation of the FEP-ANN inference rule: starting from the deterministic update above, we add zero-mean Gaussian noise directly to the post-activation state (Langevin-style)

$$\boldsymbol{\sigma}^{(t+1)} = S\left(\beta J \boldsymbol{\sigma}^{(t)}\right) + \boldsymbol{\omega}^{(t)}, \boldsymbol{\omega}^{(t)} \sim \mathcal{N}\left(0, \epsilon^2 \boldsymbol{I}\right).$$ (6)

This explicit additive Gaussian noise differs from the continuous-Bernoulli noise implied by the theoretical derivation but aligns with common Langevin formulations and was empirically robust. We use deterministic updates to identify attractors and study convergence; we use stochastic updates as a generative model of multistable dynamics.

## fcANN convergence and attractors

We investigated convergence under the deterministic update (*Equation 5*) by contrasting iterations-to-convergence of the empirical fcANN against a permutation-based null. The null was constructed by randomly permuting the upper triangle of $J$ and reflecting it to preserve symmetry (destroying topology while preserving weight distribution). For each of 1000 permutations, we initialized both models with the same random state and counted iterations to convergence. Statistical significance of faster convergence in the empirical connectome was assessed via a one-sided Wilcoxon signed-rank test on paired iteration counts (1000 pairs), testing whether the empirical connectome converges in fewer iterations than its permuted counterpart. We repeated this procedure across inverse-temperature values $\beta \in \{0.035, 0.040, 0.045, 0.050, 0.055, 0.060\}$ (yielding 2–8 attractor states). See *Figure 2—figure supplement 4* for detailed results.

## fcANN projection

We mapped out the fcANN state space by initializing the model with a random input and applying the stochastic update (*Equation 6*) for iterations, storing visited activity configurations. We performed principal component analysis (PCA) on the samples and used the first two PCs as the coordinate system for the fcANN projection. Using multinomial logistic regression, we predicted attractor identity from the first two PCs. Performance was evaluated with 10-fold cross-validation and a two-sided permutation test (1000 label permutations within folds). We visualized attractor positions and decision boundaries based on this classifier. The fcANN projection is used for visualization; unless otherwise noted, inferential comparisons between empirical and simulated data are performed in the full 122-dimensional space.

## Parameter optimization

Based on the convergence analysis, we selected the inverse temperature providing the fastest median convergence ($\beta = 0.04$) for subsequent analyses, to minimize computational costs. We then optimized the stochastic noise level $\epsilon$ for the Langevin-style update by comparing the full 122-dimensional distributions of empirical and fcANN-generated data.

Specifically, for eight $\epsilon$ values spaced logarithmically between 0.1 and 1, we generated samples with *Equation 6* and computed the 2-Wasserstein (Earth Mover's) distance between the 122-dimensional empirical and simulated distributions (after per-region z-scoring). As a null, we drew surrogate samples from a covariance-matched multivariate normal distribution $\mathcal{N}\left(0, \boldsymbol{\Sigma}\right)$, where $\boldsymbol{\Sigma}$ is the empirical

covariance of the regional time series. For each $\epsilon$, we generated 1000 null surrogates and contrasted the empirical Wasserstein distance to the null distribution, summarizing the separation with Glass's $\Delta$ and permutation p-values. We selected $\epsilon = 0.37$ for all subsequent analyses.

## Replicability

We obtained the four attractor states in study 1, as described above. We then constructed two other fcANNs, based on the study-mean functional connectome obtained in studies 2 and 3 and obtained all attractor states of these models, with the same parameter settings ($\beta = 0.04$ and $\epsilon = 0.37$) as in study 1. In both replication studies, we found four attractor states. The spatial similarity of attractor states across studies was evaluated by Pearson's correlation coefficient.

## Evaluation: resting state dynamics

To evaluate the explanatory power of the fcANN projection, we performed PCA on the preprocessed fMRI time frames from study 1 (analogous to the methodology of the fcANN projection, but on the empirical time-series data). Next, we fitted linear regression models using the first two fcANN-based or data-based PCs as regressors to reconstruct the real fMRI time frames. In-sample explained variances and confidence intervals were calculated for both models with bootstrapping (100 samples; percentile 99% CIs). Differences in explained variance between fcANN- and data-based PCs were assessed with a two-sided percentile bootstrap on the difference in $R^2$. To evaluate out-of-sample generalization of the PCs (fcANN- and data-based) from study 1, we calculated how much variance they explain in study 2.

Similarity between state occupancy and distribution was calculated during parameter optimization. More detail on the associated null models can be found in *Figure 3—figure supplement 2*.

To confirm that the real and fcANN temporal sequences (from the stochastic relaxation) display similar temporal autocorrelation properties, we compared both to their randomly shuffled variant with a 'flow analysis.' First, we calculated the direction on the fcANN projection plane between each successive TR (a vector on the fcANN projection plane for each TR transition), both for the empirical and the shuffled data. Next, we obtained two-dimensional binned means for both the x and y coordinates of these transition vectors (pooled across all participants), calculated over a 100×100 grid of uniformly distributed bins in the [−6, 6] range (arbitrary units) and applied Gaussian smoothing with $\sigma = 5$ bins (same approach as described in the Parameter optimization section). Finally, we visualized the difference between the binned-mean trajectories of the empirical and the shuffled data as a 'streamplot,' with the Python package matplotlib. The same approach was repeated with the fcANN-generated data. The similarity of the real and fcANN-generated flow analysis was quantified with Pearson's correlation coefficient (two-sided); p-values were obtained with permutation testing (1000 permutations), by shuffling temporal order (for resting-state analyses) or condition labels (for task analyses; see below).

## Evaluation: task-based dynamics

We used study 4 to evaluate the ability of the fcANN approach to capture and predict task-induced alterations in large-scale brain dynamics. First, runs 1, 3, and 7, investigating the passive experience and the down- and up-regulation of pain, respectively, were preprocessed with the same workflow used to preprocess studies 1–3 (Preprocessing and time-series extraction). Regional time series were grouped into 'pain' and 'rest' blocks, with a 6 s delay to adjust for the hemodynamic response time. All activation time frames were transformed to the fcANN projection plane obtained from study 1. Within-participant differences in the average location on the fcANN projection were calculated and visualized with radial plots, showing the participant-level mean trajectory on the projection plane from rest to pain (circles), as well as the group-level trajectory (arrow). The significance of the position difference and energy difference of the participant-level mean activations on the projection plane was tested with a two-sided permutation test (1000 permutations), using the L2 norm of the two-dimensional position difference and the absolute energy difference, respectively, as test statistics, and randomly swapping the conditions within each participant.

To further highlight the difference between the task and rest conditions, a 'flow analysis' was performed to investigate the dynamic trajectory differences between the conditions rest and pain. The analysis method was identical to the flow analysis of the resting state data (Evaluation: resting

state dynamics). First, we calculated the direction in the projection plane between each successive TR during the rest conditions (a vector on the fcANN projection plane for each TR transition). Next, we obtained two-dimensional binned means for the x and y coordinates of these transition vectors (pooled across all participants), calculated over a two-dimensional grid of 100×100 uniformly distributed bins in the [–6,6] range (arbitrary units) and applied Gaussian smoothing with $\sigma = 5$ bins. The same procedure was repeated for the pain condition, and the difference in the mean directions between the two conditions was visualized as 'streamplots' (using Python's matplotlib). We used the same approach to quantify the difference in characteristic state transition trajectories between the up- and downregulation conditions. The empirically estimated trajectory differences (from real fMRI data) were contrasted to the trajectory differences predicted by the fcANN model from study 1. The similarity between real and simulated flow maps was quantified with Pearson's correlation coefficient (two-sided), and significance was assessed via permutation testing (1000 permutations) by randomly swapping condition labels within participants.

To obtain fcANN-simulated state transitions in resting conditions, we used the stochastic relaxation procedure (*Equation 6*), with *μ* set to zero. To simulate the effect of pain-related activation on large-scale brain dynamics, we set $\mu_i$ during the stochastic relaxation procedure to a value representing pain-elicited activity in region i. The region-wise activations were obtained by calculating the parcel-level mean activations from the meta-analytic pain activation map from *Zunhammer et al., 2021*, which contained Hedges' g effect sizes from an individual participant-level meta-analysis of 20 pain studies, encompassing a total of n=603 participants. The whole activation map was scaled with five different values ranging from $10^{-3}$ to $10^{-1}$, spaced logarithmically, to investigate various signal-to-noise scenarios. We obtained the activity patterns of $10^5$ iterations from this stochastic relaxation procedure and calculated the state transition trajectories with the same approach used with the empirical data. Next, we calculated the fcANN-generated difference between the rest and pain conditions and compared it to the actual difference through a permutation test with 1000 permutations, randomly swapping the conditions within each participant in the real data and using Pearson's correlation coefficient between the real (permuted) and fcANN-generated flow maps as the test statistic. From the five investigated signal-to-noise values, we chose the one that provided the highest similarity to the real pain vs. rest trajectory difference.

When comparing the simulated and real trajectory differences between pain up- and downregulation, we used the same procedure, with two differences. First, when calculating the simulated state transition vectors for the self-regulation conditions, we used the same procedure as for the pain condition, but introduced an additional signal in the nucleus accumbens, with a negative and positive sign, for up- and downregulation, respectively. We did not optimize the signal-to-noise ratio for the nucleus accumbens signal but, instead, simply used the value optimized for the pain vs. rest contrast (For a robustness analysis, see *Figure 4—figure supplement 2*).

## Clinical data

To demonstrate the sensitivity of the fcANN approach to clinically relevant alterations of large-scale brain dynamics in autism spectrum disorder (ASD), we obtained data from n=172 individuals, acquired at the New York University Langone Medical Center (NYU) as shared in the Autism Brain Imaging Data Exchange dataset (study 7: ABIDE, *di Martino et al., 2014*). We focused on the largest ABIDE imaging center to ensure that results were not biased by center effects. We excluded high-motion cases similarly to studies 1–4, i.e., by scrubbing volumes with FD >0.15 and excluding participants with >50% of data scrubbed. Time-series data were pooled and visualized on the fcANN projection of study 1, separately for ASD and control participants. Next, for each participant, we grouped time frames from the regional time-series data according to the corresponding attractor states (obtained with the fcANN model from study 1) and averaged time frames corresponding to the same attractor state to calculate participant-level mean attractor activations. We assessed mean attractor-activity differences between groups with a two-sided permutation test, randomly reassigning group labels 50,000 times. Reported effect sizes in the clinical tables are mean activation differences. Note that activation time series were standard-scaled independently for each region, so effect size can be interpreted as the proportion of regional variability. We adjusted the significance threshold with a Bonferroni correction, accounting for tests across four states and 122 regions, resulting in α = 0.001. Finally, we calculated trajectory differences between the two groups, as predicted by the group-specific fcANNs (initialized

with the ASD and TDC connectomes), and - similarly to the approach used in study 4 - contrasted the fcANN predictions with trajectory differences observed in real rsfMRI data. As in the previous flow analyses, we tested the significance of the similarity (Pearson's correlation) between predicted and observed trajectory differences with a two-sided permutation test (1000 permutations), by shuffling group labels.

## Project website

https://pni-lab.github.io/connattractor/.

## Acknowledgements

The work was supported by the Deutsche Forschungsgemeinschaft (DFG, German Research Foundation; projects 'TRR289 - Treatment Expectation', ID 422744262 and 'SFB1280 - Extinction Learning', ID 316803389) and by IBS-R015-D1 (Institute for Basic Science; CW-W).

## Additional information

### Funding

| Funder | Grant reference number | Author |
|---|---|---|
| Deutsche Forschungsgemeinschaft | 422744262 | Ulrike Bingel Tamas Spisak |
| Deutsche Forschungsgemeinschaft | 316803389 | Ulrike Bingel Tamas Spisak |
| Institute for Basic Science | IBS-R015-D2 | Choong-Wan Woo |

The funders had no role in study design, data collection and interpretation, or the decision to submit the work for publication.

### Author contributions

Robert Englert, Resources, Data curation, Software, Validation, Investigation, Visualization, Methodology, Writing – original draft; Balint Kincses, Conceptualization, Resources, Data curation, Investigation, Writing – review and editing; Raviteja Kotikalapudi, Jialin Li, Data curation, Writing – review and editing; Giuseppe Gallitto, Resources, Writing – review and editing; Kevin Hoffschlag, Resources, Data curation, Writing – review and editing; Choong-Wan Woo, Resources, Data curation, Investigation, Writing – review and editing; Tor D Wager, Resources, Data curation; Dagmar Timmann, Investigation, Writing – review and editing; Ulrike Bingel, Funding acquisition, Investigation, Writing – review and editing; Tamas Spisak, Conceptualization, Software, Formal analysis, Supervision, Funding acquisition, Validation, Investigation, Visualization, Methodology, Writing – original draft, Writing – review and editing

### Author ORCIDs

Robert Englert https://orcid.org/0000-0002-6421-576X
Balint Kincses https://orcid.org/0000-0002-0478-6384
Raviteja Kotikalapudi https://orcid.org/0000-0003-4604-3367
Giuseppe Gallitto https://orcid.org/0000-0001-5185-0206
Jialin Li https://orcid.org/0000-0002-0310-904X
Kevin Hoffschlag https://orcid.org/0009-0000-1609-7407
Choong-Wan Woo https://orcid.org/0000-0002-7423-5422
Tor D Wager https://orcid.org/0000-0002-1936-5574
Dagmar Timmann https://orcid.org/0000-0003-1935-416X
Ulrike Bingel https://orcid.org/0000-0002-9528-3204
Tamas Spisak https://orcid.org/0000-0002-2942-0821

### Ethics

Human subjects: This work analyzed publicly available datasets.All the studies involved in the present analyses were conducted in accordance with the Declaration of Helsinki, complying with all relevant

ethical regulations for work with human participants and was approved by the local or national ethics committees. Participants in all involved studies gave written informed consent before testing. Register Numbers for studies 1-3: 4974-14, 18-8020-BO and 057617/2015/OTIG at the Ruhr University Bochum, University Hospital Essen and ETT TUKEB Hungary, respectively. Study 4 was approved by the Columbia University Institutional Review Board (Protocol number AAAE3743). Studies 5 and 6 were meta-analyses (for details see Yarkoni et al., 2011 and Zunhammer et al., 2018, 2021). Data from Study 7 was approved by the NYU-SOM IRB (https://fcon_1000.projects.nitrc.org/indi/abide/abide_ I.html).

Reviewer #1 (Public review): https://doi.org/10.7554/eLife.98725.3.sa1
Reviewer #2 (Public review): https://doi.org/10.7554/eLife.98725.3.sa2
Author response https://doi.org/10.7554/eLife.98725.3.sa3

## Additional files

### Supplementary files
MDAR checklist

### Data availability
Studies 1, 2 and 4 are available at openneuro.org (ds002608, ds002609, ds000140). Data for studies 5-6 are available at the GitHub page of the project: https://github.com/pni-lab/connattractor, copy archived at *Spisak et al., 2026*. Study 7 is available at https://fcon_1000.projects.nitrc.org/indi/ abide/ (RRID:SCR_003612), preprocessed data are available at http://preprocessed-connectomes-project.org/. Studies 1, 2 and 4 are publicly available at OpenNeuro (https://openneuro.org; accession numbers ds002608 and ds000140). Study 7 (ABIDE) is publicly available at https://fcon_1000. projects.nitrc.org/indi/abide/ (RRID:SCR_003612); preprocessed data are available at http://preprocessed-connectomes-project.org/. Meta-analytic summary data for studies 5 and 6 are available at the GitHub repository of the project: https://github.com/pni-lab/connattractor (*Spisak et al., 2026*). Data for study 3 are available upon request from the corresponding author. Analysis source code is available at https://github.com/pni-lab/connattractor (*Spisak et al., 2026*).

The following previously published datasets were used:

| Author(s) | Year | Dataset title | Dataset URL | Database and Identifier |
|---|---|---|---|---|
| Spisak T, Kincses B, Schlitt F, Zunhammer M, Schmidt-Wilcke T, Kincses ZT, Bingel U | 2020 | RPN - Study 1 | https://openneuro. org/datasets/ ds002608 | OpenNEURO, ds002608 |
| Spisak T, Kincses B, Schlitt F, Zunhammer M, Schmidt-Wilcke T, Kincses ZT, Bingel U | 2020 | RPN - Study 2 | https://openneuro. org/datasets/ ds002609 | OpenNeuro, ds002609 |
| Woo CW, Roy M, Buhle JT, Wager TD | 2018 | Distinct brain systems mediate the effects of nociceptive input and self-regulation on pain | https://openneuro. org/datasets/ ds000140 | OpenNEURO, ds000140 |

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

## Appendix 1

**Supplementary methods**

### Study 4 instructions for upregulation

'During this scan, we are going to ask you to try to imagine as hard as you can that the thermal stimulations are more painful than they are. Try to focus on how unpleasant the pain is, for instance, how strongly you would like to remove your arm from it. Pay attention to the burning, stinging, and shooting sensations. You can use your mind to turn up the dial of the pain, much like turning up the volume dial on a stereo. As you feel the pain rise in intensity, imagine it rising faster and faster and going higher and higher. Picture your skin being held up against a glowing hot metal or fire. Think of how disturbing it is to be burned, and visualize your skin sizzling, melting, and bubbling as a result of the intense heat.'

### Study 4 instructions for downregulation

'During this scan, we are going to ask you to try to imagine as hard as you can that the thermal stimulations are less painful than they are. Focus on the part of the sensation that is pleasantly warm, like a blanket on a cold day. You can use your mind to turn down the dial of your pain sensation, much like turning down the volume dial on a stereo. As you feel the stimulation rise, let it numb your arm, so any pain you feel simply fades away. Imagine your skin is very cool, from being outside, and think of how good the stimulation feels as it warms you up.'

## Appendix 2

### Derivation of the joint steady-state probability of free energy minimizing attractor networks

See (*Spisak and Friston, 2025*) for details on the whole framework.

### Regularity assumptions

- Existence of a non-equilibrium steady-state density p* with $\phi = \log p*$ differentiable almost everywhere on its support; stationary edge fluxes are finite.
- Measurable jump kernel with finite total update rate at each configuration $\sigma$; independent Poisson clocks as specified.
- Compact support of single-site updates $x \in [-1, 1]$ so boundary terms vanish in the summation-by-parts/test-function identities used for solenoidal orthogonality.
- Sufficient smoothness to justify differentiating the reversible increment at $x = \sigma_i$; mixed partials commute where invoked.

### Setup: Continuous-Bernoulli kernel

We assume that the single-site conditional distribution is the Continuous-Bernoulli on $[-1, 1]$ with canonical parameter

$$\kappa_i \left( \sigma_{-i} \right) = b_i + \sum_{j \neq i} J_{ij} \sigma_j \tag{7}$$

and density (for $x \in [-1, 1]$)

$$K \left( x \mid \kappa \right) = h \left( x \right) e^{\kappa x - A(\kappa)}, x \in [-1, 1] \tag{8}$$

with $h \left( x \right) = 1/2 \ 1_{[-1,1]} \left( x \right)$ and $A \left( \kappa \right) = \log \left( \sinh \kappa \right) - \log \kappa$ (so the $\kappa \to 0$ limit is uniform on $[-1, 1]$).
The conditional mean is:

$$L \left( \kappa \right) = E_{K(\cdot | \kappa)} \left[ x \right] = \coth \kappa - \frac{1}{\kappa} \tag{9}$$

### 1. Master equation

Let's consider a precise formalization of the asynchronous update procedure: each site $i$ has an independent Poisson clock of rate $\gamma_i$. When it rings, $\sigma_i$ is replaced by a draw $x \sim K \left( \cdot \mid \kappa_i \left( \sigma_{-i} \right) \right)$.

With $\sigma^{(i,x)}$ the configuration equal to $\sigma$ but with coordinate $i$ set to $x$, the transition density is

$$w_i \left( \sigma^{(i,x)} \mid \sigma \right) = \gamma_i K \left( x \mid \kappa_i \left( \sigma_{-i} \right) \right) \tag{10}$$

The master equation for $p \left( \sigma, t \right)$ is

$$\partial_t p(\sigma, t) = \sum_{i=1}^{N} \int_{-1}^{1} \left[ \underbrace{w_i \left( \sigma \mid \sigma^{(i,x)} \right) p \left( \sigma^{(i,x)}, t \right)}_{inflow} - \underbrace{w_i \left( \sigma^{(i,x)} \mid \sigma \right) p(\sigma, t)}_{outflow} \right] dx \tag{11}$$

**Note:** In a small interval dt, probability in state $\sigma$ changes by inflow from all one-site predecessors $\sigma^{(i,x)}$ minus outflow from $\sigma$ to those states. Multiple updates within dt are $\mathcal{O} \left( dt^2 \right)$ and negligible. Because the updated value $x$ is continuous, we integrate over $x \in [-1, 1]$.

### 2. Probability currents

Let's denote the site-wise flux density as:

$$F_i \left( \sigma \to x \right) := w_i \left( \sigma^{(i,x)} \mid \sigma \right) p^* \left( \sigma \right) \tag{12}$$

At steady state $p^*$, by definition we have:

$$\sum_i \int_{-1}^{1} \left( F_i \left( \sigma^{(i,x)} \to \sigma \right) - F_i \left( \sigma \to x \right) \right) dx = 0 \tag{13}$$

## 3. Detailed balance condition

A special (and maybe the most intuitive) way to satisfy the previous eq. is the case of detailed balance or equilibrium. In this case, every transition is balanced:

$$F_i \left( \sigma^{(i,x)} \to \sigma \right) = F_i \left( \sigma \to x \right) \tag{14}$$

It can be shown that:

$$w_i \left( \sigma^{(i,x)} \mid \sigma \right) p^* \left( \sigma \right) = w_i \left( \sigma \mid \sigma^{(i,x)} \right) p^* \left( \sigma^{(i,x)} \right) \iff \partial_{\sigma_i} \Phi \left( \sigma \right) = \kappa_i \left( \sigma_{-i} \right) = b_i + \sum_{k \neq i} J_{ik} \sigma_k \tag{15}$$

i.e., the log-density $\Phi$ changes by the local slope at $i$ (see detailed derivation below).

Hence for $j \neq i$:

$$\partial_{\sigma_j} \partial_{\sigma_i} \Phi = J_{ij}, \partial_{\sigma_i} \partial_{\sigma_j} \Phi = J_{ji} \tag{16}$$

Mixed partials commute, thus:

$$detailed\ balance \Rightarrow J_{ij} = J_{ji} \tag{17}$$

Therefore, **equilibrium (detailed balance) is possible only when the coupling is symmetric**.

**Derivation**

From:

$$w_i \left( \sigma^{(i,x)} \mid \sigma \right) p^* \left( \sigma \right) = w_i \left( \sigma \mid \sigma^{(i,x)} \right) p^* \left( \sigma^{(i,x)} \right)$$

We take logs ($\Phi = \log p^*$) and rearrange:

$$\Phi \left( \sigma^{(i,x)} \right) - \Phi \left( \sigma \right) = \log w_i \left( \sigma^{(i,x)} \mid \sigma \right) - \log w_i \left( \sigma \mid \sigma^{(i,x)} \right)$$

Substitute $w_i = \gamma_i K \left( \cdot \mid \kappa \right)$ with $\kappa = \kappa_i(\sigma_{-i}) = \kappa_i(\sigma_{-i}^{(i,x)})$:

$$\Phi \left( \sigma^{(i,x)} \right) - \Phi \left( \sigma \right) = \left[ \log \gamma_i + \log K \left( x \mid \kappa \right) \right] - \left[ \log \gamma_i + \log K \left( \sigma_i \mid \kappa \right) \right] = \log K \left( x \mid \kappa \right) - log K \left( \sigma_i \mid \kappa \right)$$

Use CB form $\log K \left( z \mid \kappa \right) = \log h \left( z \right) + \kappa z - A \left( \kappa \right)$:

$$\Phi \left( \sigma^{(i,x)} \right) - \Phi \left( \sigma \right) = \left[ \log h \left( x \right) - \log h \left( \sigma_i \right) \right] + \kappa \left( x - \sigma_i \right)$$

($A \left( \kappa \right)$ cancels because $\kappa$ is the same).

For Continuous–Bernoulli on $[-1,1]$, $h \left( x \right) = 1/2$ on the support $\Rightarrow \log h \left( x \right) - \log h \left( \sigma_i \right) = 0$, so:

$$\Phi \left( \sigma^{(i,x)} \right) - \Phi \left( \sigma \right) = \kappa \left( x - \sigma_i \right) \tag{18}$$

Differentiating at $x = \sigma_i$ yields

$$\partial_{\sigma_i} \Phi \left( \sigma \right) = \kappa_i \left( \sigma_{-i} \right) = b_i + \sum_{k \neq i} J_{ik} \sigma_k \tag{19}$$

## 4. Non-equilibrium steady state (NESS)

Detailed balance (equilibrium) is only one specific way the steady-state condition can hold:

$$\sum_i \int_{-1}^{1} \left( F_i\left(\sigma^{(i,x)} \to \sigma\right) - F_i\left(\sigma \to x\right) \right) dx = 0 \tag{20}$$

Let's consider the general case: an arbitrary (possibly asymmetric) J coupling. J can always be decomposed into:

$$J = J^{\mathrm{S}} + J^{\mathrm{A}}, J^{\mathrm{S}} := \frac{1}{2}\left(J + J^{\top}\right), J^{\mathrm{A}} := \frac{1}{2}\left(J - J^{\top}\right) \tag{21}$$

This leads to a decomposition of the edge currents on each directed edge $\left(\sigma, \sigma^{(i,x)}\right)$:

$$F_i^{\mathrm{sym}}(\sigma \to x) := \frac{1}{2}\left(F_i(\sigma \to x) + F_i(\sigma^{(i,x)} \to \sigma)\right), \qquad F_i^{\mathrm{sol}}(\sigma \to x) := \frac{1}{2}\left(F_i(\sigma \to x) - F_i(\sigma^{(i,x)} \to \sigma)\right). \tag{22}$$

so that $F_i = F_i^{\mathrm{sym}} + F_i^{\mathrm{sol}}$ and $F_i^{\mathrm{sym}}(\sigma \to x) = F_i^{\mathrm{sym}}\left(\sigma^{(i,x)} \to \sigma\right)$.

Since $F_i(\sigma \to x) = e^{\Phi(\sigma)} w_i(\sigma \to x)$, define rates accordingly:

$$w_i^{\mathrm{sym}}(\sigma \to x) := e^{-\Phi(\sigma)} F_i^{\mathrm{sym}}(\sigma \to x), w_i^{\mathrm{sol}}(\sigma \to x) := e^{-\Phi(\sigma)} F_i^{\mathrm{sol}}(\sigma \to x) \tag{23}$$

so $w_i = w_i^{\mathrm{sym}} + w_i^{\mathrm{sol}}$ and $F_i^{\mathrm{sym}} = e^{\Phi} w_i^{\mathrm{sym}}, F_i^{\mathrm{sol}} = e^{\Phi} w_i^{\mathrm{sol}}$.

Note that this circulating flow is not generally tangent to the level sets of $\phi$ unless the mobility-noise (fluctuation-dissipation) conditions of the continuous theory apply [*Ao, 2004*].

## Appendix 3

Here, we provide a concise derivation of how $\sigma$ and J changes as a consequence of free energy minimization resulting in an inference (node update) rule and a local, incremental learning rule, respectively. For more background and a detailed derivation, see *Spisak and Friston, 2025*.

Let the instantaneous net input to node $\sigma_i$ be

$$u_i := b_i + \sum_{j \neq i} J_{ij} \sigma_j, \tag{31}$$

and let $q(\sigma_i) \propto e^{b_q \sigma_i}$ be the variational marginal with mean $L(b_q)$, where $L$ is the Langevin function (the mean of the Continuous-Bernoulli).

### Inference (Hopfield-style)

We start by writing up the local variational free energy from the point of view of a single node $\sigma_i$:

$$F = E_q\left[\ln q(\sigma_i)\right] - E_q\left[\ln p(\sigma_{\searrow i} \mid \sigma_i)\right] \tag{32}$$

Next, we express:

$$\ln q(\sigma_i) = b_q \sigma_i - A(b_q) + \ln h(\sigma_i) \tag{33}$$

with mean $E_q[\sigma_i] = L(b_q) = A'(b_q)$.
and

$$\ln p(\sigma_{\searrow i} \mid \sigma_i) = \text{const} + \sigma_i \sum_{j \neq i} J_{ij} \sigma_j. \tag{34}$$

As it depends on $\sigma_i$ only through the linear slope $\sum_{j \neq i} J_{ij} \sigma_j$:

$$E_q\left[\ln p(\sigma_{\searrow i} \mid \sigma_i)\right] = \text{const} + L(b_q) \sum_{j \neq i} J_{ij} \sigma_j \tag{35}$$

Now, we assemble the local free energy (dropping constants independent of $b_q$):

$$F = (b_q - b_i) L(b_q) - [A(b_q) A(b_i)] - L(b_q) \sum_{j \neq i} J_{ij} \sigma_j. \tag{36}$$

Equivalently, using $u_i = b_i + \sum_{j \neq i} J_{ij} \sigma_j$ and $A'(b_q) = L(b_q)$,

$$F = (b_q - u_i) L(b_q) - A(b_q) + \text{const.} \tag{37}$$

1. Differentiate w.r.t. $b_q$ and use $A'(b_q) = L(b_q)$ to cancel terms:

$$\frac{\partial F}{\partial b_q} = (b_q - u_i) L'(b_q) + L(b_q) - A'(b_q) = (b_q - u_i) L'(b_q). \tag{38}$$

Setting the derivative to zero gives $b_q^\star = u_i$ and, therefore, the node-wise update

$$E_q[\sigma_i] = L(b_q^\star) = L\left(b_i + \sum_{j \neq i} J_{ij} \sigma_j\right) \tag{39}$$

which is the stochastic Hopfield/Boltzmann-style activation with the Langevin nonlinearity.

### Learning (Hebb–anti-Hebb)

Make the dependence on $u_i$ explicit by adding and subtracting the CB log-partition $A(u_i)$ (using $\ln p(\sigma_i \mid \sigma_{\searrow i}) = u_i \sigma_i - A(u_i) + \ln h(\sigma_i)$ and Bayes' rule):

$$F = E_q \left[ (b_q - u_i) \, \sigma_i \right] + A \left( u_i \right) - A \left( b_q \right) + \text{const.} \tag{40}$$

Now

$$\frac{\partial F}{\partial u_i} = -E_q \left[ \sigma_i \right] + A' \left( u_i \right) = -E_q \left[ \sigma_i \right] + L \left( u_i \right), \tag{41}$$

and by the chain rule with $\partial u_i / \partial J_{ij} = \sigma_j$ we obtain

$$\frac{\partial F}{\partial J_{ij}} = \left[ L \left( u_i \right) - E_q \left[ \sigma_i \right] \right] \sigma_j. \tag{42}$$

Using a stochastic (sample-based) estimate $E_q \left[ \sigma_i \right] \approx \sigma_i$ and descending $F$ gives the local update

$$\Delta J_{ij} \propto \underbrace{\sigma_i \sigma_j}_{\text{Hebbian (observed)}} \underbrace{- L \left( b_i + \sum_{k \neq i} J_{ik} \sigma_k \right) \sigma_j}_{\text{anti-Hebbian (predicted)}} \tag{43}$$

This is a predictive coding-based learning rule that strengthens observed correlations and subtracts those already predicted by the current model.

## Appendix 4

### Self-orthogonalization of attractor states

To illustrate how the learning rule in free energy minimizing attractor networks gives rise to efficient, (approximately) orthogonal representations of the external states, suppose the network has already learned a pattern $s^{(1)}$, whose neural representation is the attractor $\sigma^{(1)}$ and associated weights $J^{(1)}$. When a new pattern $s^{(2)}$ is presented that is correlated with $s^{(1)}$, the network's prediction for $\sigma_i^{(2)}$ will be $\hat{\sigma}_i = L\left(b_i + \sum_{k \neq i} J_{ik}^{(1)} \sigma_k\right)$. Because inference with $J^{(1)}$ converges to $\sigma^{(1)}$ and $\sigma^{(2)}$ is correlated with $\sigma^{(1)}$, the prediction $\sigma$ will capture variance in $\sigma^{(2)}$ that is 'explained' by $\sigma^{(1)}$. The learning rule updates the weights based only on the unexplained (residual) component of the variance, the prediction error. In other words, $\sigma$ approximates the projection of $\sigma^{(2)}$ onto the subspace already spanned by $\sigma^{(1)}$. Therefore, the weight update primarily strengthens weights corresponding to the component of $\sigma^{(2)}$ that is orthogonal to $\sigma^{(1)}$. Thus, the learning effectively encodes this residual, $\sigma_\perp^{(2)}$, ensuring that the new attractor components being formed tend to be orthogonal to those already established. For more details on self-orthogonalization in these networks, including an empirical demonstration, see *Spisak and Friston, 2025*.

## Appendix 5

### Reconstruction of the attractor network from the activation timeseries

We start from *Equation 1*, written in matrix notation:

$$\mathrm{E_{HN}}(\sigma) = -\tfrac{1}{2}\sigma^\top \mathrm{J}\sigma + \sigma^\top \mathrm{b} = -\tfrac{1}{2}\sigma^\top \mathrm{J}\sigma + \sigma^\top \mathrm{JJ}^{-1}\mathrm{b} = -\tfrac{1}{2}\sigma^\top \mathrm{J}\sigma + \sigma^\top \mathrm{JJ}^{-1}\mathrm{b} - \tfrac{1}{2}\mathrm{b}^\top \mathrm{J}^{-1}\mathrm{b} + \tfrac{1}{2}\mathrm{b}^\top \mathrm{J}^{-1}\mathrm{b}$$

To complete the square, we added and subtracted $\tfrac{1}{2}b^\top J^{-1}b$ within the exponent. We recognize that $\sigma^\top JJ^{-1}b = \sigma^\top b$. Now add and subtract $\tfrac{1}{2}\mathrm{b}^\top \mathrm{J}^{-1}\mathrm{b} = -\tfrac{1}{2}\left[\sigma^\top \mathrm{J}\sigma - 2\sigma^\top \mathrm{JJ}^{-1}\mathrm{b} + \mathrm{b}^\top \mathrm{J}^{-1}\mathrm{b}\right] + \tfrac{1}{2}\mathrm{b}^\top \mathrm{J}^{-1}\mathrm{b} = -\tfrac{1}{2}\left[\sigma - \mathrm{J}^{-1}\mathrm{b}\right]^\top \mathrm{J}\left[\sigma - \mathrm{J}^{-1}\mathrm{b}\right] + \tfrac{1}{2}\mathrm{b}^\top \mathrm{J}^{-1}\mathrm{b}$

$$= -\tfrac{1}{2}\left[\sigma^\top \mathrm{J}\sigma - 2\sigma^\top \mathrm{JJ}^{-1}\mathrm{b} + \mathrm{b}^\top \mathrm{J}^{-1}\mathrm{b}\right] + \tfrac{1}{2}\mathrm{b}^\top \mathrm{J}^{-1}\mathrm{b} = -\tfrac{1}{2}\left[\sigma - \mathrm{J}^{-1}\mathrm{b}\right]^\top \mathrm{J}\left[\sigma - \mathrm{J}^{-1}\mathrm{b}\right] + \tfrac{1}{2}\mathrm{b}^\mathrm{T}\mathrm{J}^{-1}\mathrm{b}$$

Given that $P(\sigma) = \exp\left(-E_{HN}(\sigma)\right)$, the expression simplifies to:

$$\mathrm{P}(\sigma) \propto \exp\left(-\tfrac{1}{2}\left(\sigma - \mathrm{J}^{-1}\mathrm{b}\right)^\top \mathrm{J}\left(\sigma - \mathrm{J}^{-1}\mathrm{b}\right)\right)$$

Note that the term $\tfrac{1}{2}b^\top J^{-1}b$ is independent of $\sigma$; we have absorbed it into the normalization constant. This is exactly the exponent of a multivariate Gaussian distribution with mean $J^{-1}b$ and covariance matrix $J^{-1}$, meaning that the weight matrix of the attractor network can be reconstructed as the inverse covariance matrix of activation timeseries of the lower-level nodes: $J = -\Lambda = -\Sigma^{-1}$

