## [Editor Report · eLife Assessment]

This study presents a **valuable** approach for revealing large-scale brain attractor dynamics during resting states, task processing, and disease conditions using insights from Hopfield neural networks. The evidence supporting the findings is **convincing** across the many datasets analysed. The work will be of broad interest to neuroscientists using neuroimaging data with interest in computational modelling of brain activity.

---

## [Referee Report · Reviewer #1 (Public review)]

Summary:

Englert et al. proposed a functional connectivity-based Attractor Neural Network (fcANN) to reveal attractor states and activity flows across various conditions, including resting state, task-evoked, and pathological conditions. The large-scale brain attractors reconstructed by fcANNs are orthogonal organization, which is in line with the free-energy theoretical framework. Additionally, the fcANN demonstrates differences in attractor states between individuals with autism and typically developing individuals.

The study used seven datasets, which ensures robust replication and validation of generalization across various conditions. The study is a representative example that combines experimental evidence based on fcANN and the theoretical framework. The fcANN projection offers an interesting way of visualization, allowing researchers to observe attractor states and activity flow patterns directly. Overall, the study may offer valuable insights into brain dynamics and computational neuroscience.

Comments on revision:

The authors have addressed my previous concerns and substantially improved the manuscript. Fig.4 and Fig.5 still keep fcHNN rather than the updated fcANN.

---

## [Referee Report · Reviewer #2 (Public review)]

Summary:

Englert et al. use a novel modelling approach called functional connectome-based Hopfield Neural Networks (fcHNN) to describe spontaneous and task-evoked brain activity, and the alterations in brain disorders. Given its novelty, the authors first validate the model parameters (the temperature and noise) with empirical resting-state function data and against null models. Through the optimisation of the temperature parameter, they first show that the optimal number of attractor states is four before fixing the optimal noise that best reflects the empirical data, through stochastic relaxation. Then, they demonstrate how these fcHNN generated dynamics predict task-based functional activity relating to pain and self-regulation. To do so, they characterise the different brain states (here as different conditions of the experimental pain paradigm) in terms of the distribution of the data on the fcHNN projections and flow-analysis. Lastly, a similar analysis was performed on a population with autism condition. Through Hopfield modeling, this work proposes a comprehensive framework that links various types of functional activity under a unified interpretation with high predictive validity.

Strengths:

The phenomenological nature of the Hopfield model and its validation across multiple datasets presents a comprehensive and intuitive framework for the analysis of functional activity. The results presented in this work further motivate the study of phenomenological models as an adequate mechanistic characterisation of large-scale brain activity.

Following up from Cole et al. 2016, the authors put forward a hypothesis that many of the changes to the brain activity, here, in terms of task-evoked and clinical data, can be inferred from the resting-state brain data alone. This brings together neatly the idea of different facets of brain activity emerging from a common space of functional (ghost) attractors.

The use of the null models motivates the benefit for non-linear dynamics in the context of phenomenological models when assessing the similarity to the real empirical data.

Comments on revision:

I am happy with how the authors addressed the comments and am happy to move ahead without further comments.

---

## [Author Response]

The following is the authors’ response to the original reviews.

**Public Reviews:**

**Reviewer #1 (Public Review):**
Summary:Englert et al. proposed a functional connectome-based Hopfield artificial neural network (fcHNN) architecture to reveal attractor states and activity flows across various conditions, including resting state, task-evoked, and pathological conditions. The fcHNN can reconstruct characteristics of resting-state and task-evoked brain activities. Additionally, the fcHNN demonstrates differences in attractor states between individuals with autism and typically developing individuals.Strengths:(1) The study used seven datasets, which somewhat ensures robust replication and validation of generalization across various conditions.(2) The proposed fcHNN improves upon existing activity flow models by mimicking artificial neural networks, thereby enhancing the representational ability of the model. This advancement enables the model to more accurately reconstruct the dynamic characteristics of brain activity.(3) The fcHNN projection offers an interesting visualization, allowing researchers to observe attractor states and activity flow patterns directly.

We are grateful to the reviewer for highlighting the robustness of our findings across multiple datasets and for appreciating the novelty and representational advantages of our fcHNN model (which has been renamed to fcANN in the revised manuscript).

Weaknesses:(1) The fcHNN projection can offer low-dimensional dynamic visualizations, but its interpretability is limited, making it difficult to make strong claims based on these projections. The interpretability should be enhanced in the results and discussion.

We thank the reviewer for these important points. We agree that the interpretability of the low-dimensional projection is limited. In the revised manuscript, we have reframed the fcANN projection primarily as a visualization tool (see e.g. line 359) and moved the corresponding part of Figure 2 to the Supplementary Material (Supplementary Figure 2). We have also implemented a substantial revision of the manuscript, which now directly links our analysis to the novel theoretical framework of self-orthogonalizing attractor networks (Spisak & Friston, 2025), opening several new avenues in terms of interpretation and shedding light on the computational principles underlying attractor dynamics in the brain (see the revised introduction and the new section “Theoretical background”, starting at lines 128, but also the Mathematical Appendices 1-2 in the Supplementary Material for a comprehensive formal derivation). As part of these efforts, we now provide evidence for the brain’s functional organization approximating a special, computationally efficient class of attractor networks, the so-called Kanter-Sompolinsky projector network (Figure 2A-C, line 346, see also our answer to your next comment). This is exactly, what the theoretical framework of free-energy-minimizing attractor networks predicts.

(2) The presentation of results is not clear enough, including figures, wording, and statistical analysis, which contributes to the overall difficulty in understanding the manuscript. This lack of clarity in presenting key findings can obscure the insights that the study aims to convey, making it challenging for readers to fully grasp the implications and significance of the research.

We have thoroughly revised the manuscript for clarity in wording, figures (see e.g. lines 257, 482, 529 in the Results and lines 1128, 1266, 1300, 1367 in the Methods). We carefully improved statistical reporting and ensured that we always report test statistics, effect sizes and clearly refer to the null modelling approach used (e.g. lines 461, 542, 550, 565, 573, 619, as well as Figures 2-4). As absolute effect sizes, in many analyses, do not have a straightforward interpretation, we provided Glass’ , as a standardized effect size measure, expressing the distance of the true observation from the null distribution as a ratio of the null standard deviation. To further improve clarity, we now clearly define our research questions and the corresponding analyses and null models in the revised manuscript, both in the main text and in two new tables (Tables 1 and 2). We denoted research questions and null model with Q1-7 and NM1-5, respectively and refer to them at multiple instances when detailing the analyses and the results.

**Reviewer #2 (Public Review):**
Summary:Englert et al. use a novel modelling approach called functional connectome-based Hopfield Neural Networks (fcHNN) to describe spontaneous and task-evoked brain activity and the alterations in brain disorders. Given its novelty, the authors first validate the model parameters (the temperature and noise) with empirical resting-state function data and against null models. Through the optimisation of the temperature parameter, they first show that the optimal number of attractor states is four before fixing the optimal noise that best reflects the empirical data, through stochastic relaxation. Then, they demonstrate how these fcHNN-generated dynamics predict task-based functional activity relating to pain and self-regulation. To do so, they characterise the different brain states (here as different conditions of the experimental pain paradigm) in terms of the distribution of the data on the fcHNN projections and flow analysis. Lastly, a similar analysis was performed on a population with autism condition. Through Hopfield modeling, this work proposes a comprehensive framework that links various types of functional activity under a unified interpretation with high predictive validity.Strengths:The phenomenological nature of the Hopfield model and its validation across multiple datasets presents a comprehensive and intuitive framework for the analysis of functional activity. The results presented in this work further motivate the study of phenomenological models as an adequate mechanistic characterisation of large-scale brain activity.Following up on Cole et al. 2016, the authors put forward a hypothesis that many of the changes to the brain activity, here, in terms of task-evoked and clinical data, can be inferred from the resting-state brain data alone. This brings together neatly the idea of different facets of brain activity emerging from a common space of functional (ghost) attractors.The use of the null models motivates the benefit of non-linear dynamics in the context of phenomenological models when assessing the similarity to the real empirical data.

We thank the reviewer for recognizing the comprehensive and intuitive nature of our framework and for acknowledging the strength of our hypothesis that diverse brain activity facets emerge from a common resting state attractor landscape.

Weaknesses:While the use of the Hopfield model is neat and very well presented, it still begs the question of why to use the functional connectome (as derived by activity flow analysis from Cole et al. 2016). Deriving the functional connectome on the resting-state data that are then used for the analysis reads as circular.

We agree that starting from functional couplings to study dynamics is in stark contrast with the common practice of estimating the interregional couplings based on structural connectome data. We now explicitly discuss how this affects the scope of the questions we can address with the approach, with explicit notes on the inability of this approach to study the structure-function coupling and its limitations in deriving mechanistic insights at the level of biophysical implementation.

Line 894:

“The proposed approach is not without limitations. First, as the proposed approach does not incorporate information about anatomical connectivity and does not explitly model biophysical details. Thus, in its present form, the model is not suitable to study the structure-function coupling and cannot yiled mechanistic explanations underlying (altered) polysynaptic connections, at the level of biophysical details.”

We are confident, however, that our approach is not circular. At the high level, our approach can be considered as a function-to-function generative model, with twofold aims.

First, we link large-scale brain dynamics to theoretical artificial neural network models and show that the functional connectome display characteristics that render it as an exceptionally “well-behaving” attractor network (e.g. superior convergence properties, as contrasted against appropriate respective null models). In the revised manuscript, we have significantly improved upon this aspect by explicitly linking the fcANN model to the theoretical framework of self-orthogonalizing attractor networks (Spisak & Friston, 2025) (see the revised introduction and the new section “Theoretical background”, starting at lines 128, but also the Mathematical Appendices 1-2 in the Supplementary Material for a comprehensive formal derivation). As part of these efforts, we now provide evidence for the brain’s functional organization approximating a special, computationally efficient class of attractor networks, the so-called Kanter-Sompolinsky projector network (Figure 2A-C, line 346, see also our answer to your next comment). This is exactly, what the theoretical framework of free-energy-minimizing attractor networks predicts. This result is not circular, as the empirical model does not use the key mechanism (the Hebbian/anti-Hebbian learning rule) that induces self-orthogonalization in the theoretical framework. We clarify this in the revised manuscript, e.g. in line 736.

Second, we benchmark ability of the proposed function-to-function generative model to predict unseen data (new datasets) or data characteristics that are not directly encompassed in the connectivity matrix (e.g. non-Gaussian conditional dependencies, temporal autocorrelation, dynamical responses to perturbations on the system). These benchmarks are constructed against well defined null models, which provide reasonable references. We have now significantly improved the discussion of these null models in the revised manuscript (Tables 1 and 2, lines 257). We not only show, that our model - when reconstructing resting state dynamics - can generalize to unseen data over and beyond what is possible with the baseline descriptive measure (e.g. covariance measures and PCA), but also demonstrate the ability of the framework to reconstruct the effects of perturbations on this dynamics (such as task-evoked changes), based solely on the resting state data from another sample.

If the fcHNN derives the basins of four attractors that reflect the first two principal components of functional connectivity, it perhaps suffices to use the empirically derived components alone and project the task and clinical data on it without the need for the fcHNN framework.

We are thankful for the reviewer for highlighting this important point, which encouraged us to develop a detailed understanding of the origins of the close alignment between attractors and principal components (eigenvectors of the coupling matrix) and the corresponding (approximate) orthogonality. Here, we would like to emphasize that the attractor-eigenvector correspondence is by no means a general feature of any arbitrary attractor network. In fact, such networks are a very special class of attractor neural networks (the so-called Kanter-Sompolinsky projector neural network, Kanter & Sompolinsky, 1987), with a high degree of computational efficiency, maximal memory capacity and perfect memory recall. It has been rigorously shown that in such networks, the eigenvectors of the coupling matrix (i.e. PCA on the timeseries data) and the attractors become equivalent (Kanter & Sompolinsky, 1987). This in turn made us ask the question, what are the learning and plasticity rules that drive attractor networks towards developing approximately orthogonal attractors? We found that this is a general tendency of networks obeying the free energy principle (Figure 2A-C, line 346, see also our answer to your next comment). The formal derivation of this framework in now presented in an accompanying theoretical piece (Spisak & Friston, 2025). In the revised manuscript, we provide a short, high-level overview of these results (in the Introduction form line 55 and in the new section “Theoretical background”, line 128, but also the Mathematical Appendices 1-2 in the Supplementary Material for a comprehensive formal derivation). According to this new theoretical model, attractor states can be understood as a set of priors (in the Bayesian sense) that together constitute an optimal orthogonal basis, equipping the update process (which is akin to a Markov-chain Monte Carlo sampling) to find posteriors that generalize effectively within the spanned subspace. Thus, in sum, understanding brain function in terms of attractor dynamics - instead of PCA-like descriptive projections - provides important links towards a Bayesian interpretation of brain activity. At the same time, the eigenvector-attractor correspondence also explains, why descriptive decomposition approaches, like PCA or ICA are so effective at capturing the dynamics of the system, at the first place.

As presented here, the Hopfield model is excellent in its simplicity and power, and it seems suited to tackle the structure-function relationship with the power of going further to explain task-evoked and clinical data. The work could be strengthened if that was taken into consideration. As such the model would not suffer from circularity problems and it would be possible to claim its mechanistic properties. Furthermore, as mentioned above, in the current setup, the connectivity matrix is based on statistical properties of functional activity amongst regions, and as such it is difficult to talk about a certain mechanism. This contention has for example been addressed in the Cole et al. 2016 paper with the use of a biophysical model linking structure and function, thus strengthening the mechanistic claim of the work.

We agree that investigating how the structural connectome constraints macro-scale dynamics is a crucial next step. Linking our results with the theoretical framework of self-orthogonalizing attractor networks provides a principled approach to this, as the “self-orthogonalizing” learning rule in the accompanying theoretical work provides the opportunity to fit attractor networks with structural constraints to functional data, shedding light on the plastic processes which maintain the observed approximate orthogonality even in the presence of these structural constraints. We have revised the manuscript to clarify that our phenomenological approach is inherently limited in its ability to answer mechanistic questions at the level of biophysical details (lines 894) and discuss this promising direction as follows:

Lines 803:

“A promising application of this is to consider structural brain connectivity (as measured by diffusion MRI) as a sparsity constraint for the coupling weights and then train the fcANN model to match the observed resting-state brain dynamics. If the resulting structural-functional ANN model is able to closely match the observed functional brain substate dynamics, it can be used as a novel approach to quantify and understand the structural functional coupling in the brain”.

**Recommendations for the authors:**

**Reviewer #1 (Recommendations For The Authors):**
(1) The statistical analyses are poorly described throughout the manuscript. The authors should provide more details on the statistical methods used for each comparison, as well as the corresponding statistics and degrees of freedom, rather than solely reporting p-values.

We thank the reviewer for pointing this out. We have revised the manuscript to include the specific test statistics, precise p-values and raw effect sizes for all reported analyses to ensure full transparency and replicability, see e.g. lines 461, 542, 550, 565, 573, 619, as well as Figures 2-4. Additionally, as absolute effect sizes - in many analyses - do not have a straightforward interpretation, we provided Glass’ Δ, as a standardized effect size measure, expressing the distance of the true observation from the null distribution as a ratio of the null standard deviation.

We have also improved the description of the statistical methods used in the manuscript (lines 1270, 1306, 1339, 1367, 1404) and added two overview tables (Tables 1 and 2) that summarize the methodological approaches and the corresponding null models.

Furthermore, we have fully revised the analysis corresponding to noise optimization. We only retained null model 2 (covariance-matched Gaussian) in the main text and on Figure 3, and moved model 1 (spatial phase randomization) into the Supplementary Material (Supplementary Figure 6) and is less appropriate for this analysis (trivially significant in all cases). Furthermore, as test statistic, no we use a Wasserstein distance between the 122-dimensional empirical and the simulated data (instead of focusing on the 2-dimensional projection). This analysis now directly quantifies the capacity of the fcANN model to capture non-Gaussian conditionals in the data.

(2) The convergence procedure is not clearly explained in the manuscript. Is this an optimization procedure to minimize energy? If so, the authors should provide more details about the optimizer used.

We apologize for the lack of clarity. The convergence is not an optimization procedure per se, in a sense that it does not involve any external optimizer. It is simply the repeated (deterministic) application of the same update rule also known from Hopfield networks or Boltzmann machines. However, as detailed in the accompanying theoretical paper, this update rule (or inference rule) inherently solves and optimization problem: it performs gradient descent on the free energy landscape of the network. As such, it is guaranteed to converge to a local free energy minimum in the deterministic case. We have clarified this process in the Results and Methods sections as follows:

Line 161:

“Inference arises from minimizing free energy with respect to the states \sigma. For a single unit, this yields a local update rule homologous to the relaxation dynamics in Hopfield networks”.

Line 181:

“In the basis framework (Spisak & Friston, 2025), inference is a gradient descent on the variational free energy landscape with respect to the states σ and can be interpreted as a form of approximate Bayesian inference, where the expected value of the state σ_i_ is interpreted as the posterior mean given the attractor states currently encoded in the network (serving as a macro-scale prior) and the previous state, including external inputs (serving as likelihood in the Bayesian sense)”.

Line 1252:

“As the inference rule was derived as a gradient descent on free energy, iterations monotonically decrease the free energy function and therefore converge to a local free‑energy minimum without any external optimizer. Thus, convergence does not require any optimization procedure with an external optimizer. Instead, it arises as the fixed point of repeated local inference updates, which implement gradient descent on free energy in the deterministic symmetric case.”

(3) In Figure 2G, the beta values range from 0.035 to 0.06, but they are reported as 0.4 in the main text and the Supplementary Figure. Please clarify this discrepancy.

We are grateful to the reviewer for spotting this typo. The correct value for β is 0.04, as reported in the Methods section. We have corrected this inconsistency in the revised manuscript and as well as in Supplementary Figure 5.

(4) Line 174: What type of null model was used to evaluate the impact of the beta values? The authors did not provide details on this anywhere in the manuscript.

We apologize for this omission. The null model is based on permuting the connectome weights while retaining the matrix symmetry, which destroys the specific topological structure but preserves the overall weight distribution. We have now clarified this at multiple places in the revised manuscript (lines 432, Table 1-2, Figure 2), and added new overview tables (Tables 1 and 2) to summarize the methodological approaches and the corresponding null models.

(5) Figure 3B: It appears that the authors only demonstrate the reproducibility of the “internal” attractor across different datasets. What about other states?

Thank you for noticing this. We now visualize all attractor states in Figure 3B (note that these essentially consist of two symmetric pairs).

(6) Figure 3: What does “empirical” represent in Figure 3? Is it PCA? If the “empirical” method, which is a much simpler method, can achieve results similar to those of the fcHNN in terms of state occupancy, distribution, and activity flow, what are the benefits of the proposed method? Furthermore, the authors claim that the explanatory power of the fcHNN is higher than that of the empirical model and shows significant differences. However, from my perspective, this difference is not substantial (37.0% vs. 39.9%). What does this signify, particularly in comparison to PCA?

This is a crucial point that is now a central theme of our revised manuscript. The reviewer is correct that the “empirical” method is PCA. PCA - by identifying variance-heavy orthogonal directions - aims to explain the highest amount of variance possible in the data (with the assumption of Gaussian conditionals). While empirical attractors are closely aligned to the PCs (i.e. eigenvectors of the inverse covariance matrix, as shown in the new analysis Q1), the alignment is only approximate. We basically take advantage of this small “gap” to quantify, weather attractor states are a better fit to the unseen data than the PCs. Obviously, due to the otherwise strong PC-attractor correspondence, this is expected to be only a small improvement. However, it is an important piece of evidence for the validity of our framework, as it shows that attractors are not just a complementary, perhaps “noisier” variety of the PCs, but a “substrate” that generalizes better to unseen data than the PCs themselves. We have revised the manuscript to clarify this point (lines 528).

**Reviewer #2 (Recommendations For The Authors):**
For clarity, it might be useful to define and use consistently certain key terms. Connectome often refers to structural (anatomical) connectivity unless defined specifically this should be considered, in Figure 1B title for example Brain state often refers to different conditions ie autism, neurotypical, sleep, etc... see for review Kringelbach et al. 2020, Cell Reports. When referring to attractors of brain activity they might be called substates.

We thank the reviewer for these helpful suggestions. We have carefully revised the manuscript to ensure our terminology is precise and consistent. We now explicitly refer to the “functional connectome” (including the title) and avoid using the too general term “brain state” and use “substates” instead.

In Figure 2 some terms are not defined. Noise is sigma in the text but elpsilon in the figure. Only in methods, the link becomes clear. Perhaps define epsilon in the caption for clarity. The same applies to μ in the methods. It is only described above in the methods, I suggest repeating the epsilon definition for clarity

We appreciate this feedback and apologize for the inconsistency. We have revised all figures and the Methods section to ensure that all mathematical symbols (including ε, σ, and μ) are clearly and consistently defined upon their first appearance and in all figure captions. For instance, noise level is now consistently referred to as ϵ. We improved the consistency and clarity for other terms, too, including:

functional connectome-based Hopfiled network (fcHNN) => functional connectivity-based attractor network (fcANN);

temperature => inverse temperature;

And improved grammar and language throughout the manuscript.

References

Kanter, I., & Sompolinsky, H. (1987). Associative recall of memory without errors. Physical Review A, 35(1), 380–392. 10.1103/physreva.35.380

Spisak T & Friston K (2025). Self-orthogonalizing attractor neural networks emerging from the free energy principle. arXiv preprint arXiv:2505.22749.